# CoIn dual-atom catalyst for hydrogen peroxide production via oxygen reduction reaction in acid

Jiannan Du[1,4], Guokang Han ●[1,4] ✉, Wei Zhang[1], Lingfeng Li[1], Yuqi Yan ●[1], Yaoxuan Shi[1], Xue Zhang ●[2], Lin Geng[3], Zhijiang Wang[1], Yueping Xiong[1], Geping Yin ●[1] & Chunyu Du ●[1] ✉

The two-electron oxygen reduction reaction in acid is highly attractive to produce $H_2O_2$, a commodity chemical vital in various industry and household scenarios, which is still hindered by the sluggish reaction kinetics. Herein, both density function theory calculation and in-situ characterization demonstrate that in dual-atom CoIn catalyst, O-affinitive In atom triggers the favorable and stable adsorption of hydroxyl, which effectively optimizes the adsorption of OOH on neighboring Co. As a result, the oxygen reduction on Co atoms shifts to two-electron pathway for efficient $H_2O_2$ production in acid. The $H_2O_2$ partial current density reaches 1.92 mA cm$^{-2}$ at 0.65 V in the rotating ring-disk electrode test, while the $H_2O_2$ production rate is as high as 9.68 mol g$^{-1}$ h$^{-1}$ in the three-phase flow cell. Additionally, the CoIn-N-C presents excellent stability during the long-term operation, verifying the practicability of the CoIn-N-C catalyst. This work provides inspiring insights into the rational design of active catalysts for $H_2O_2$ production and other catalytic systems.

Hydrogen peroxide ($H_2O_2$) is an important green oxidant with wide range applications in both industries and household scenarios, including pulp/textile bleaching, waste-water treatment, chemical synthesis and disinfection[1–3]. Nevertheless, the traditional anthraquinone-based methods for $H_2O_2$ production are energy intensive, and are not environmental-friendly as a large amount of toxic organic solvents are required[4,5]. The electrochemical two-electron oxygen reduction reaction (2e-ORR) marks a promising route for clean $H_2O_2$ production, due to its mild aqueous condition and capability of being powered by green electricity, which is completely compatible with the sustainable economy. In particular, the electrochemical $H_2O_2$ production in acid media is the most desirable because $H_2O_2$ is stable and more oxidative in the low pH region[5].

The energy conversion efficiency of acidic $H_2O_2$ electrolyzers is mainly determined by the cathodic 2e-ORR catalysts. The catalysts with high activity and $H_2O_2$ selectivity enable a large $H_2O_2$ production

rate at a minimum overpotential, which is critical to the practical application of acidic $H_2O_2$ electrolyzers[6–8]. However, the present developed catalysts, including the precious metal catalysts such as PtHg/PtAu alloys and the non-precious metal carbon-based catalysts, cannot achieve satisfying activity and selectivity for the acidic 2e-ORR[8–10]. Moreover, increasing polarization of the cathode catalysts even worsens their selectivity to $H_2O_2$, further lowering the overall energy conversion efficiency of $H_2O_2$ electrolyzers[5].

Recently, single-atom catalysts (SACs) have shown excellent 2e-ORR activity and selectivity in alkaline media, mainly due to their atomically dispersed metal centers such as Ni, Co, Pt, Pd, and Mo[11–17]. Inspired by these studies, a few attempts have thus been made to catalyze the 2e-ORR using SACs in acid media. Unfortunately, the acidic 2e-ORR kinetics on SACs is far slower than that in alkaline solution, despite the optimization of various metal coordination configurations[11,18]. Moreover, the acidic ORR on SACs often follows the

[1]School of Chemistry and Chemical Engineering, Harbin Institute of Technology, Harbin 150001, PR China. [2]Center for Materials and Interfaces, Shenzhen Institutes of Advanced Technology, Chinese Academy of Sciences, Shenzhen 518055, PR China. [3]School of Materials Science and Engineering, Harbin Institute of Technology, Harbin 150001, PR China. [4]These authors contributed equally: Jiannan Du, Guokang Han. ✉e-mail: gkhan@hit.edu.cn; cydu@hit.edu.cn

four-electron pathway[19–22]. In comparison with SACs, the neighboring metal pairs within a short range in dual-atom catalysts (DACs) enable more possibilities of tuning the adsorption properties of reaction intermediates[23–25], and might provide an effective method of boosting the 2e-ORR activity and selectivity.

Herein, a CoIn-N-C dual-atom catalyst (DAC) is proposed as an effective 2e-ORR catalyst for $H_2O_2$ production in acid media. The density function theory (DFT) calculations reveal that the valance electron number, as well as the d-band center of Co 3d orbital, can be regulated by OH-blocked In, which optimizes the bonding of key OOH intermediate on Co. This moderate adsorption of OOH grants CoIn-N-C the favorable 2e-ORR kinetics, reaching the apex of the volcano-type plot between predicted activity and OOH adsorption energy. Electrochemical kinetics analysis demonstrates that the rate constant of ORR via 2e pathway is a magnitude higher than that via 4e pathway on CoIn-N-C, leading to a $H_2O_2$ yield of >90% in a wide potential range. The partial current density for $H_2O_2$ production (i$H_2O_2$), a criterion for evaluating the $H_2O_2$ productivity, is as high as 1.92 mA cm$^{-2}$ at 0.65 V for the CoIn-N-C DAC. When evaluated in the three-phase-flow cell, a $H_2O_2$ production rate of 9.68 mol g$^{-1}$ h$^{-1}$ is achieved at the current density of 100 mA cm$^{-2}$. Our work demonstrates the possibility of manipulating the reaction kinetics and pathway of ORR by the unique electronic interaction of DACs, providing inspiring insights into the rational design of the active sites for the 2e-ORR with targeted activity and selectivity.

## Results and discussion

Since the Co-based SACs are demonstrated to have better 2e-ORR activity than other transition metal-based ones, we choose Co as one metal center to design the 2e-ORR DACs. Guided by the high binding capability of p-block metal elements with O as reported earlier[26], we introduce p-block metals ($M_p$, represented by Al, Ga and In from the same group) into a N-coordinated single-atom Co model coupled with a short-ranged vacancy nearby (denoted as s-CoVac) as illustrated in Fig. 1a. These proposed dual-atom models, denoted as d-CoMp, and the reference single-atom models are shown in Fig. S1. When validating the stable adsorption site for the ORR-related species on the proposed DAC models, the density function theory (DFT) calculation reveals that OOH and OH adsorb favorably on $M_p$ atoms (Fig. S2) compared with Co by the much distinctive adsorption energies (Table S1). In particular, OH is strongly bonded on $M_p$ atoms (Fig. 1b) and remains stable within the working potential range for $H_2O_2$ electrolyzers (Fig. S3). Thus, it is believed that the actual structures for d-CoMp models in the working environment are OH-blocked[22,27], as illustrated by the inset in Fig. 1b, which are denoted as d-CoMpOH for further analysis.

After blocked by OH, the $M_p$ atoms in d-CoMpOH become less O-affinitive, making Co the favorable adsorption site for O-containing species (Fig. S4 and Table S1). Due to the presence of $M_p$OH moieties, the local charge distribution of d-CoMpOH models rearranges compared to s-CoVac, which is indicated by the charge density difference after introducing MpOH (Figs. 1c, and S5). Especially, the charge density around Co atom increases, demonstrating the electron donating nature of MpOH moieties. This enriched electron density around Co not only increases the valance electron number (Table S2), but regulates the 3d orbital structure, causing the positive shift of d-band center as revealed by the partial density of states (pDOS) in Fig. 1d. Due to the strong dependence of intermediates adsorption on the electronic structure of catalytic active site, it is believed that the adsorption of key intermediate OOH for 2e-ORR on Co atom in the three d-CoMpOH models is greatly altered.

Crystal orbital Hamilton population (COHP) analysis was then performed to look into the bonding nature between Co and O from OOH. As revealed by Fig. 1e, upon introduction of $M_p$OH, the anti-bonding interaction below Fermi level between Co and O decreases for all the three d-CoMpOH models, suggesting the strengthened OOH

bonding. It is found that both the integrated COHP below Fermi level (iCOHP) and the OOH adsorption energy ($\Delta G_{OOH}$) are negatively relevant to the valance electron number of Co (Fig. 1f), which is the strong evidence of the adsorption properties regulated by the electronic structure.

The limiting potential ($U_L$), representing the potential below which the 2e-ORR become an exothermic reaction, is predicted on the proposed models. Apparently, d-CoInOH is located at the apex of the volcano-type plot between the limiting potential and the adsorption energy of OOH for 2e-ORR (Fig. 1g and Table S3), due to its moderate bonding strength to OOH induced by the proper electron transfer from InOH moiety. In comparison, excessive electrons are donated to Co atoms from AlOH and GaOH moieties, leading to relatively strong adsorption of OOH. In the Gibbs free energy diagram for the 2e-ORR at the equilibrium potential of 0.7 V (Fig. 1h), the energy differences further demonstrate a low overpotential of 48 mV on d-CoInOH, much lower than that on the single-atom Co-based (s-CoVac and s-Co, the traditional CoN$_4$ SAC without vacancy) and other reference Co-based dual atom models.

Guided by the DFT prediction, the CoIn-N-C DAC was experimentally prepared, as illustrated in Fig. S6. Briefly, the CoIn co-doped zeolitic imidazolate frameworks 8 precursor (CoIn-ZIF8) was firstly synthesized, which was subsequently treated by a facile high-temperature pyrolysis process to obtain the CoIn-N-C catalyst (see Methods for details). As indicated by the scanning electron microscopy (SEM) and transmission electron microscopy (TEM) images in Fig. 2a, b, the CoIn-N-C catalyst shows a uniform rhombic dodecahedron shape, which well maintains the morphology of the CoIn-ZIF8 precursor (Fig. S7). The weak selected-area electron diffraction (SAED) pattern (inset of Fig. 2b) and the X-ray diffraction (XRD) pattern (Fig. S8a) demonstrate the poor crystalline carbon nature of the CoIn-N-C catalyst. The $N_2$ adsorption−desorption characterization substantiates the mesoporous structure of CoIn-N-C (Fig. S8b), benefitting the exposure and utilization of active site during catalysis[28,29]. The energy-dispersive X-ray spectroscopy (EDS) mapping (Fig. 2c) presents the highly homogeneous distribution of Co, In, N and C elements, suggesting that Co and In are highly dispersive throughout the material. The mass ratio of Co and In was determined to be 0.9 and 1.0 wt.% by ICP-OES, respectively (Table S4). The aberration-corrected high-angle annular dark field scanning transmission electron microscopy (AC-HAADF-STEM) image with sub-angstrom resolution (Fig. 2d) further elucidates the atomic dispersion of the two metal elements with obvious dual-metal pairs (marked with red circles). A representative isolated dual-metal pair labeled as 1 in Fig. 2d is further characterized by the Z-contrast analysis, which is a powerful tool to qualitatively identify the presence of hetero-nuclear dual-metal pairs. As shown in Fig. 2e, due to the Z-contrast difference between Co ($Z = 27$) and In ($Z = 49$), the two bright specks with different intensities confirm the presence of atomic Co-In pair in Region 1, consistent to the intensity feature described in previous reports[30,31]. Similar analysis performed on other regions (Fig. S9) suggests the common presence of Co-In atomic pair. The average distance between the two hetero-nuclear atoms is measured to be ca. 2.72 Å (Figs. 2f and S9).

To determine the accurate electronic structure and detailed local coordination environment of CoIn-N-C catalyst at the atomic scale, the synchrotron radiation-based X-ray absorption near-edge structure (XANES) and extended X-ray absorption fine structure (EXAFS) were measured. As shown in the Co K-edge XANES spectra (Figs. 3a, and S10a), the white line intensity of CoIn-N-C as well as the edge energy is located between the reference materials of Co foil and cobalt oxide (CoO), indicating an oxidation state of Co between 0 and +2, which is the characteristic for atomically dispersed metal species[32,33]. In Co-In pairs, the oxidation state of Co is lower than that in single-atom Co-N-C (Fig. S11), which indicates the electron transfer to Co driven by electronegativity. In the Fourier transformed (FT) R-spaced EXAFS spectra

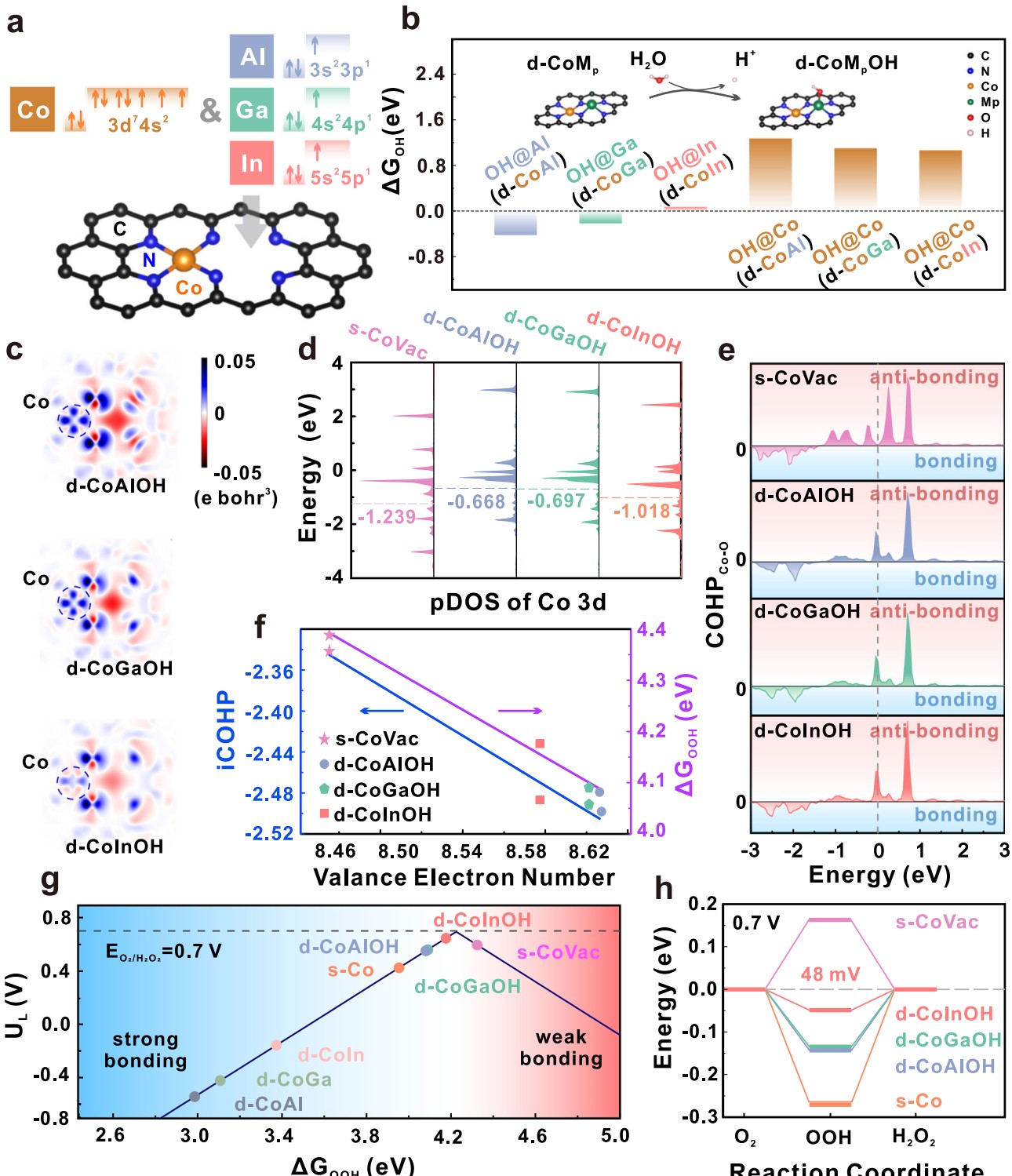

**Fig. 1 | DFT calculation and prediction of dual-atom models for 2e-ORR.** **a** Illustration of three d-CoMp models consisting of Co and p-block metals, **b** adsorption energy of OH ($\Delta G_{OH}$) on p-block metal atoms (left) and Co (right) for d-CoMp (inset: OH-blocked d-CoMpOH models), **c** cross-section of the charge density difference diagrams of d-CoMpOH models, **d** partial density of states (pDOS) and corresponding calculated d-band centers of Co 3d for d-CoMpOH and s-CoVac models, **e** crystal orbital Hamilton population (COHP) of Co from d-CoMpOH models and O from adsorbed OOH, **f** integrated COHP below Fermi level (iCOHP) and OOH adsorption energy ($\Delta G_{OOH}$) as a function of Co valence electron number, **g** volcano-type plot between limiting potential ($U_L$) and OOH adsorption energy for 2e-ORR of d-CoMpOH, d-CoMp, s-Co and s-CoVac, and **h** Gibbs free energy diagram for 2e-ORR at 0.7 V.

(Fig. 3b), a main peak similar to cobalt(II) phthalocyanine (CoPc) can be found at around 1.4 Å, which is attributed to the first shell Co-N coordination scattering[34,35]. Meanwhile, a small peak at 2.4 Å can be observed, longer than the Co-Co path at 2.15 Å for metallic Co,

suggesting the presence of a hetero-nuclear metal-metal interaction in the second path, i.e., the paired Co-In diatomic configuration[30,36]. These two coordination paths can be better visualized by the two maxima in the wavelet transform (WT) of the EXAFS (Fig. 3c), which is

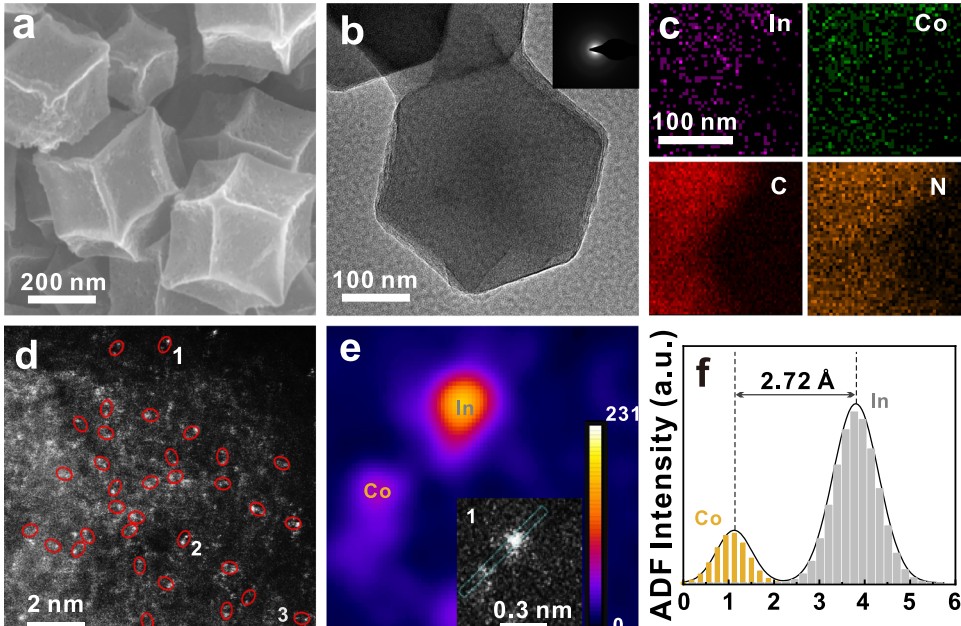

**Fig. 2 | Microscopical characterization of CoIn-N-C. a** Scanning electron microscopy (SEM) image, **b** transmission electron microscopy (TEM) image (inset: selected-area electron diffraction SAED pattern), **c** energy-dispersive X-ray spectroscopy (EDS) elemental mapping and **d** aberration-corrected high-angle annular dark field scanning transmission electron microscopy (AC-HAADF-STEM) image of CoIn-N-C, **e** colored raster graphic of region 1 in Fig. 2d and **f** corresponding intensity profile. Note that a.u. represents arbitrary units.

apparently different from the reference samples (Fig. S10a, b). Similar analysis was performed on the In spectra of CoIn-N-C and reference samples (In foil and $In_2O_3$). The oxidation state of In in CoIn-N-C is determined to be between 0 and +3 (Figs. 3d, and S10b), and the major peak at around 1.70 Å in the FT R-spaced EXAFS spectra in Fig. 3e can be identified as the dominant coordination of In with light elements[14,37]. The features of the maxima in the WT of In EXAFS (Fig. 3f) are distinguishably different from that of In and $In_2O_3$ (Fig. S12c, d), excluding their presence, and the minor maximum further suggests the possible existence of hetero-nuclear metal-metal interaction in CoIn-N-C.

The least-square EXAFS fitting was performed on both Co and In spectra to unravel the coordination structure of metal centers. As shown in Fig. 3g, the Co K-edge EXAFS spectrum can be well fitted by the Co-N and Co-In scattering paths with the coordination numbers of 4.2 and 1.1, respectively (Table S5). Meanwhile, the R space curve of In can also be fitted with the In-N and In-Co paths, with the coordination number of 3.7 and 0.7. respectively (Fig. 3h and Table S5). The above analyses indicate a local structure of Co-In diatomic sites in the form of $CoInN_6$, whose optimized configuration is shown in the Fig. 3i. It is worth mentioning that the Co-In bond length is 2.72 Å, well consistent with the length measured in HAADF-STEM images in Fig. 2f.

In situ surface-enhanced Raman scattering spectroscopy (SERS) was utilized to gain evidence on the OH adsorption on the In atoms when exposed to electrolyte. As can be seen in Figs. S13a and S14, when immersed to $N_2$-saturated $HClO_4$ electrolyte, a noticeable peak at ca. 1080 $cm^{-1}$ arises in the Raman spectra of CoIn-N-C, which corresponds to the metal-OH, as reported in previous literatures[38,39]. Due to the fact that similar peak can only be found in In-N-C, it is inferred that it is In atoms that adsorb OH in the electrolyte, which agrees well with the strong adsorption of OH on In atoms revealed by our calculation results. Isotope experiments further suggest that the adsorbed OH groups originate from $H_2O$ (Fig. S14b).

The ORR performance of CoIn-N-C in 0.1 mol $L^{-1}$ $HClO_4$ solution was firstly analyzed using the rotating ring-disk electrode (RRDE) technique (Fig. S13b) in reference to the Co-N-C and In-N-C SACs. It is

clear from Fig. 4a that the noticeable current response on the ring electrode for the CoIn-N-C DAC begins at above 0.7 V vs. reversible hydrogen electrode (RHE). Its calculated $H_2O_2$ yield (Fig. 4b) reaches 94% at 0.6 V, and remains highly selective in the wide potential range of 0.2 V-0.7 V. The electron transfer numbers calculated from the current profile recorded in both RRDE with pre-calibrated collection efficiency (Figs. 4b, and S15a, b) and RDE with different rotating rates (Fig. S15c, d) are within the range of 2.0 to 2.3, suggesting a major 2e pathway. Considering the equilibrium potential of $O_2/H_2O_2$ (0.7 V), the introduction of In transforms the ORR pathway on Co atom from 4e with moderate performance for Co-N-C to 2e with considerable reactivity for CoIn-N-C. Basic physical characterization in Fig. S16 suggests that the presence of Co-In pairs is responsible for the enhanced $H_2O_2$ selectivity and further DFT calculation indicates that only short-ranged Co-In interaction is capable of enhancing $H_2O_2$ electrosynthesis activity (Fig. S17).

Further kinetic analysis was performed using the Damjanovic model to decouple the three coupled reactions involved in ORR: direct reduction of $O_2$ to water, reduction of $O_2$ to peroxide and the $H_2O_2$ reduction reaction (HPRR)[40], as illustrated in the inset of Fig. 4c. It is clear that the rate constant for oxygen reduction to $H_2O_2$ ($k_2$) is an order of magnitude higher than that to $H_2O$ ($k_1$), suggesting fast $H_2O_2$ production kinetics on CoIn-N-C. Negligible HPRR activity is confirmed by both minimum $k_3$ and current response for $H_2O_2$ reduction (Fig. 4d). Due to the inert HPRR activity, the continuous accumulation of $H_2O_2$ generated can be achieved for more practical application[41], which will be demonstrated later. Our CoIn-N-C is one of the most potential candidates for $H_2O_2$ production in acid media with balanced selectivity and reactivity, as revealed by the remarkable partial current density for $H_2O_2$ production of 1.92 mA $cm^{-2}$ at 0.65 V which is even comparable to that in alkaline media (Fig. 4e and Table S6). Moreover, the CoIn-N-C is able to remain stable in continuous cyclic voltammetry for 20000 cycles at high potential range of 0.6 V and 1.0 V in oxygen-saturated acid solution, as neither the current response nor the $H_2O_2$ yield changes obviously after cycling (Fig. S18). The durability of CoIn-N-C is attributed to the stability of Co-In pairs. As revealed in Fig. S19, the

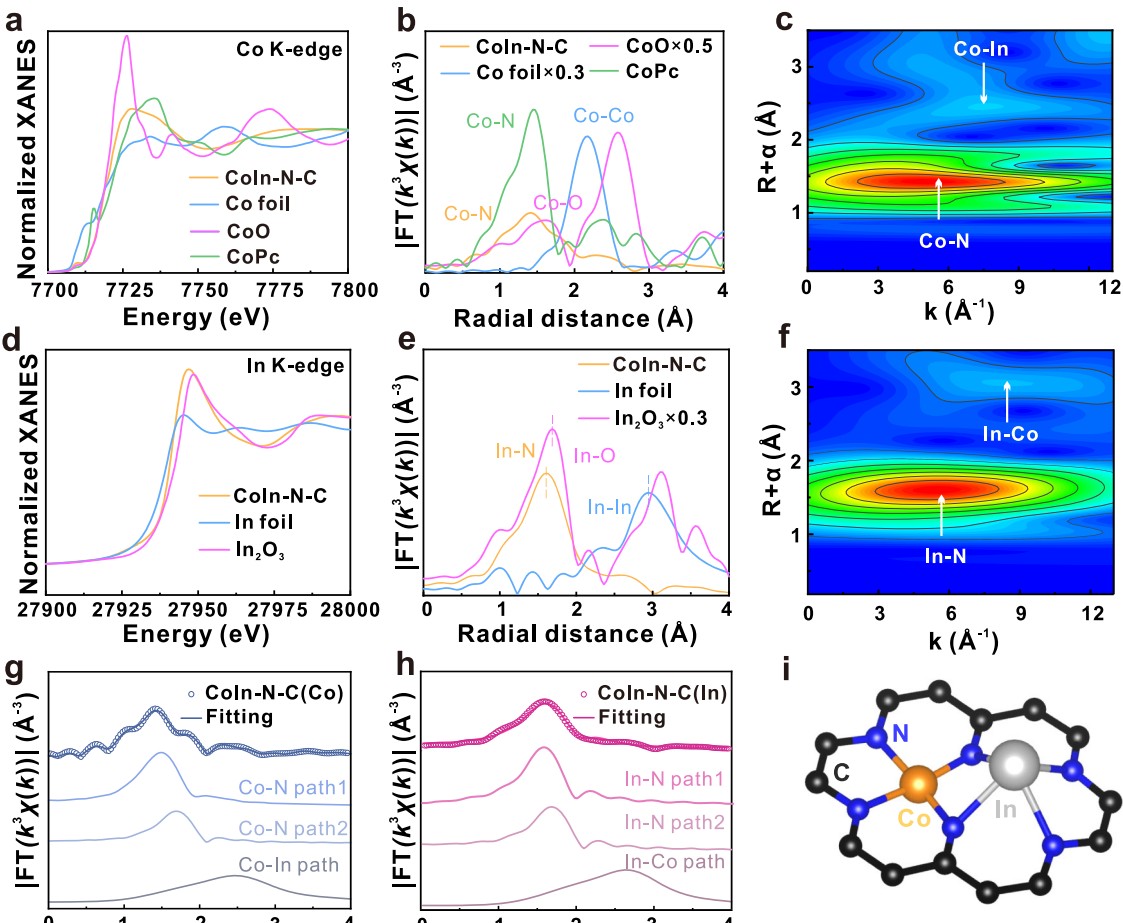

**Fig. 3 | X-ray absorption spectroscopic characterization of CoIn-N-C. a** Co K-edge X-ray absorption near-edge structure (XANES), **b** Fourier-transform extended X-ray absorption fine structure (EXAFS) spectra of CoIn-N-C and other reference samples, **c** wavelet transform (WT) contour plots of EXAFS of CoIn-N-C. **d** In K-edge XANES, **e** Fourier-transform EXAFS spectra of CoIn-N-C and other reference samples, **f** WT contour plots of EXAFS of CoIn-N-C. EXAFS fitting curves of **g** Co K-edge and **h** In K-edge EXAFS of CoIn-N-C, **i** illustration of the coordination structure of CoIn-N-C.

Co-In dual pairs can still be visualized after ADTs, and the stability of short-ranged Co-In pairs is further proved by the highest dementalization energy calculated by DFT (Fig. S20).

Due to the presence of hetero-nuclear dual-metal pair, identifying active site on DACs is challenging. As probed by the SCN⁻ poisoning experiments, both CoIn-N-C and Co-N-C can be easily poisoned by SCN⁻ and experience dramatic activity loss (Fig. S21). As revealed by the calculated adsorption energy of SCN⁻ summarized in Table S7, the adsorption of SCN⁻ on Co atoms in both s-Co and d-CoInOH models are strong with obvious Co-S bonding (Fig. S22), while In atom in d-CoInOH is not the suitable bonding site. Thus, it is believed that Co is the adsorption and reduction site of $O_2$ in CoIn-N-C, similar as in Co-N-C, which is consistent with the above DFT prediction. The reaction mechanism on CoIn-N-C can then be uncovered, as illustrated in Fig. 4f. When exposed to electrolyte, the O-affinitive In atom adsorbs OH favorably, and the $O_2$ adsorption occurs on Co site followed by the two protonation steps to produce $H_2O_2$. Under such circumstance, the InOH moiety serves as the electronic modifier to Co, optimizing the adsorption of OOH to a balanced degree where the thermodynamic barrier of 2e-ORR is minimized.

Inspired by the 2e-ORR performance in RRDE, the $H_2O_2$ electrosynthesis was studied on a practical three-phase flow cell by chronopotentiometry, with the CoIn-N-C-coated commercial gas-diffusion layer (GDL) as the working electrode (Figs. 5a, and S23a). The $H_2O_2$ concentration in the electrolyte from the cathodic tank was analyzed after 15 min electrolysis for the calculation of $H_2O_2$ production rate and

Faradic efficiency (FE). Traditional $Ce(SO_4)_2$ titration was performed to determine the $H_2O_2$ concentration, with the well fitted calibration curve for UV-vis spectrophotometric determination of $Ce^{4+}$ in aqueous solution shown in Fig. S24. It is revealed that the $H_2O_2$ production rate (9.68 mol g⁻¹ h⁻¹) with optimal FE can be achieved at 100 mA cm⁻² (Figs. 5b, and S25), comparable with that reported in base solution (Table S8).

For the $H_2O_2$ accumulation test, when operated at 100 mA cm⁻² for 4 h, 11.4 mmol $H_2O_2$ was produced with stable electrode potential (Fig. 5c), corresponding to a practical concentration of 1.938 g L⁻¹ in the electrolyte from the cathodic tank. The average $H_2O_2$ production rate is 9.22 mol g⁻¹ h⁻¹, demonstrating the stability of CoIn-N-C even in accumulated $H_2O_2$ during the long-term operation and the concentrated electrolyte. The acid electrolyte after electrolyzing for only 30 min can be utilized directly as Fenton regent with additive $Fe^{2+}$, which can effectively oxidize rhodamine B and methylene blue within 15 min (see Methods and Fig. 5d–f), demonstrating the practicability of generated $H_2O_2$ in waste-water treatment and bleaching. In a more practical two electrode system (Fig. S23b), the origin of polarization was analyzed at different operation current densities (Fig. 5g, see Methods for details). It is inspiring to find that the cathode potential of CoIn-N-C at the large current density of 100 mA cm⁻² (ca. 0.14 A) is significantly higher than that of In-N-C (Fig. S26), demonstrating the practicability of CoIn-N-C (Fig. S27).

In this work, guided by the DFT screening, we demonstrate the design and construction of a dual-atom catalyst (DAC) consisted of Co

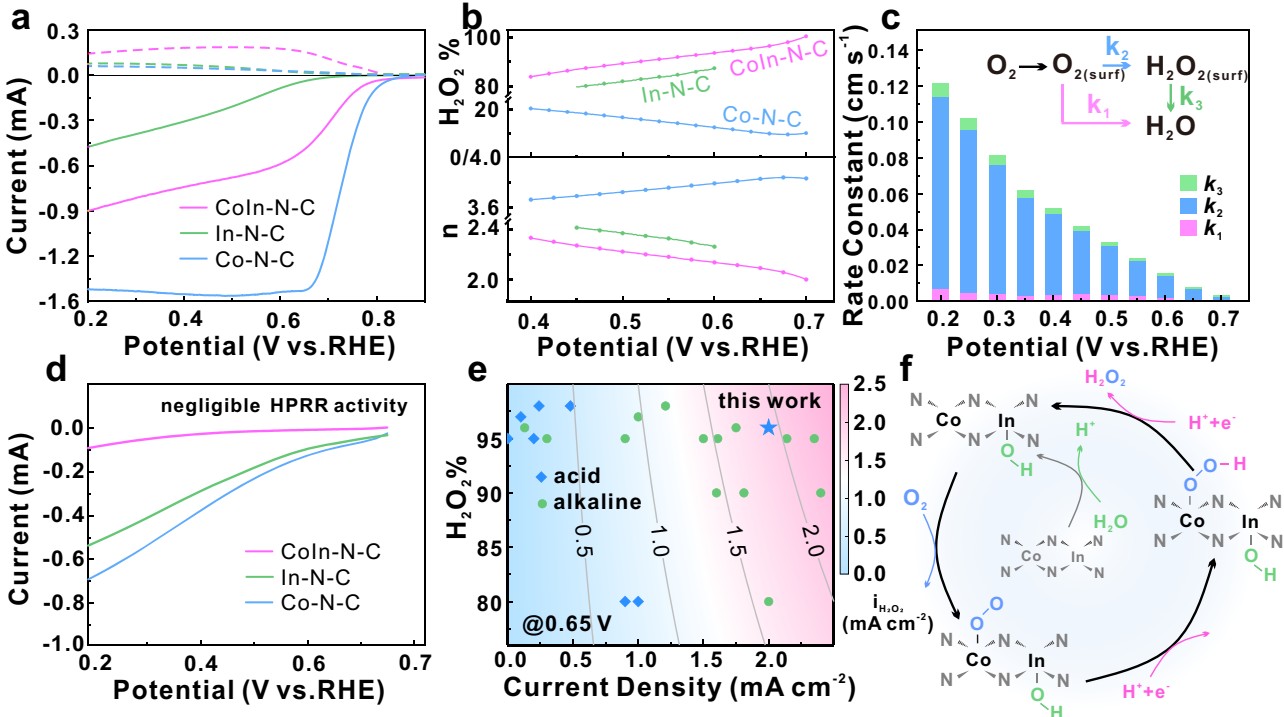

**Fig. 4 | Electrochemical performance in rotating ring-disk electrode (RRDE) and catalytic mechanism analysis. a** Oxygen reduction polarization curves (solid) and $H_2O_2$ oxidation curves (dashed) of CoIn-N-C, Co-N-C and In-N-C collected at 1600 rpm by RRDE, and **b** corresponding $H_2O_2$ selectivity and electron transfer number. **c** Rate constants of the three reactions coupled in ORR obtained by kinetics analysis using the Damjanovic model, **d** hydrogen peroxide reduction polarization curves of CoIn-N-C, Co-N-C and In-N-C, **e** comparison of RRDE performance for $H_2O_2$ production with reported catalysts in acid (blue dots) and alkaline (green dots) media, **f** illustration of the proposed 2e-ORR mechanism on CoIn-N-C.

and In metal centers and its promising application in catalyzing $O_2$ reduction to $H_2O_2$ in acid media. It is revealed that the favorable bonding of OH on In induces the appropriate regulation of Co-3d orbital structure, optimizing the adsorption of reaction intermediates. Thus, the CoIn-N-C DAC presents a $H_2O_2$ partial current density of 1.92 mA cm$^{-2}$ at 0.65 V. Moreover, the CoIn-N-C cathode enables a fast peroxide production rate of 9.68 mol g$^{-1}$ h$^{-1}$ in the three-phase flow cell operated at a large current density of 100 mA cm$^{-2}$. In addition, the CoIn-N-C DAC shows satisfying stability during the long-term operation. Our work demonstrates the possibility of effectively manipulating the reaction kinetics of ORR using the electronic interaction within DACs, providing inspiring insights into the rational design of the active sites for other catalytic systems.

## Methods

### Synthesis of CoIn-ZIF8, Co-ZIF8 and In-ZIF8

Briefly, 1.365 g (5 mmol) Co(NO$_3$)$_2$ 6H$_2$O, 2.79 g (5 mmol) Zn(NO$_3$)$_2$ 6H$_2$O and 0.159 g (0.5 mmol) In(NO$_3$)$_3$ were dissolved in 100 mL methanol. The methanol solution of 2-methylimidazole (3.08 g in 50 mL) solution was added into the Co-Zn-In solution and mixed under vigorous stirring for 15 min, which was then allowed to stand for 24 h at 25 °C. The purple solid (CoIn-ZIF8) was obtained after washing with methanol for several times by centrifugation and dried at 60 °C in vacuum for 12 h. The synthetic process of Co-ZIF8 and In-ZIF8 was similar to CoIn-ZIF8 except the usage of In and Co salts, respectively.

### Synthesis of CoIn-N-C, Co-N-C and In-N-C

The precursor powder (CoIn-ZIF8, Co-ZIF8 or In-ZIF8) was placed in a tube furnace and heated to 950 °C for 3 h at the heating rate of 5 °C min$^{-1}$ under flowing 10% H$_2$/Ar. After naturally cooled to room temperature, the catalyst powder was obtained after acid wash in 1 mol L$^{-1}$ HCl.

## Physical characterization

X-ray diffraction (XRD) analysis was performed on a Rigaku D/max 2500 diffractometer equipped with Cu-Kα radiation (k = 1.5406 Å, 40 kV, 20 mA). Morphologies of these samples and element mapping were observed on a field emission scanning electron microscope (FE-SEM, Hitachi S-4800) and a transmission electron microscope (FEI Tecnai G2 F20 S-Twin operated at 200 kV). The high-resolution spherical aberration-corrected HAADF-STEM images were recorded by JEOL JEM-ARM200F TEM/STEM with a spherical aberration corrector working at 300 kV. The N$_2$ adsorption-desorption analysis was performed on a Beishide 3H-2000PS2 analyzer. XAFS spectra were measured at the beamline BL14W1 station of the Shanghai Synchrotron Radiation Facility. The Co K-edge XANES data were recorded in transmission mode with Co foil, CoO and CoP as references. The In K-edge XAS of CoIn-N-C were recorded in fluorescence mode, with In foil and In$_2$O$_3$ as references. The acquired EXAFS data were processed according to the standard procedures using the Athena and Artemis implemented in the IFEFFIT software packages[42]. The EXAFS spectra were obtained by subtracting the post-edge background from the overall absorption and then normalizing with respect to the edge-jump step. To obtain the quantitative structural parameters around central atoms, the least-squares curve parameter fitting was performed using the ARTEMIS module of IFEFFIT software packages. Surface-enhanced Raman spectroscopy (SERS) were performed on a Renishaw inVia Reflex 03040405 Raman spectrometer using 632.8 nm laser. Airtight three-electrode flow cell with sapphire window was used for in situ Raman measurements, and multiple scans were accumulated for better signal noise ratio. The signal was enhanced using shell-isolated nanoparticle-enhanced Raman spectroscopy (SHINERS) technique[43], First Au nanoparticles (NPs) was synthesized as follows: 1.4 mL of 1 wt% sodium citrate solution was added into 200 mL of 0.01 wt% boiling HAuCl$_4$ solution under continuous stirring for 40 min. After cooling to

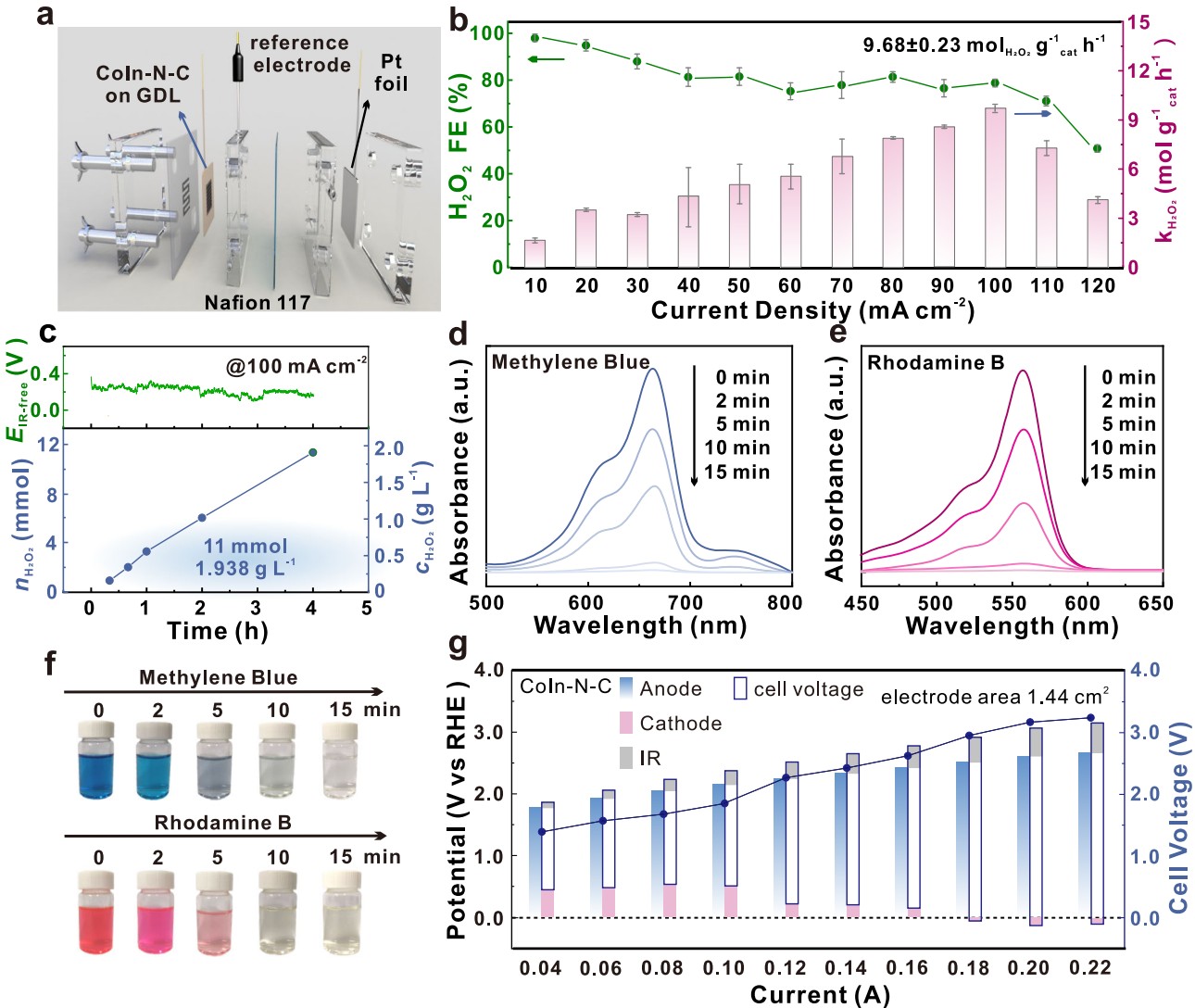

**Fig. 5 | $H_2O_2$ production performance in the flow cells and application in the dye decomposition. a** Scheme of the three-electrode flow cell setup for $H_2O_2$ production. **b** $H_2O_2$ production rate ($k_{H2O2}$) and corresponding faradaic efficiency ($H_2O_2$ FE) of the flow cell operated under different current density. The error bar represents the standard error of three independent tests. **c** The IR-corrected electrode potential ($E_{IR-free}$) of the flow cell operated under a current density of 100 mA cm$^{-2}$, where the resistance of 18.8 ± 0.3 Ω was 100% compensated, and the accumulate amount ($n_{H2O2}$) and concentration of $H_2O_2$ ($c_{H2O2}$) in the cathodic tank. UV-vis spectra during the decomposition process of **d** methylene blue and **e** rhodamine, **f** digital photographs for the decomposition process; **g** Origin of the polarization in two-electrode flow cells, where the resistance of 2.20 ± 0.15 Ω was 100% compensated. Note that a.u. represents arbitrary units.

25 °C, 0.6 mL 1 mmol L$^{-1}$ (3-Aminopropyl) trimethoxysilane solution was added to 25 mL Au NPs under stirring, then 3.2 mL sodium silicate solution was added. After 3 minutes the solution was transferred to 98 °C bath and react for 20 min to form the shell-isolated nanoparticles (SHINs) solution. Prepared SHINs was centrifuged three times for Raman measurements. The production of $H_2$ during $H_2O_2$ electro-synthesis in flow cell is evaluated by gas chromatography from the gas outlet.

**Electrochemical tests in RRDE**

All the electrochemical tests were performed in a three-electrode system using a CHI 760E electrochemistry station (CH Instrument, Shanghai). A rotating ring-disk electrode (RRDE) loaded with catalysts, a platinum sheet (1 cm$^2$), and reversible hydrogen electrode (0.1 mol L$^{-1}$ HClO$_4$) were used as the working electrode, counter electrode, and reference electrode, respectively. The RRDE consists of a glassy-carbon rotation disk electrode (disk area: 0.2475 cm$^2$) and a Pt ring. The catalyst ink was prepared by adding 2.6 mg catalyst powder into 30 μL Nafion (5 wt%, DuPont) and 520 μL mixture of water (18.25 MΩ

cm, Milli-Q) and isopropanol (ACS grade, Aladdin Reagent) in 3:1 volume to form a homogenous ink assisted by ultrasound[44]. The working electrode for RDE test is the catalyst-loaded glassy-carbon rotating disk electrode (RDE, area: 0.07065 cm$^2$). The loading of catalysts is controlled as 0.1 mg cm$^{-2}$.

The ORR polarization curves were conducted in O$_2$-saturated 0.1 mol L$^{-1}$ HClO$_4$ using cyclic voltammetry in the potential range from 1 V to 0 V (vs. RHE) at a scan rate of 10 mV s$^{-1}$ for the disk electrode. For RRDE test, a constant potential of 1.2 V was applied to the Pt ring during the test. The $H_2O_2$ yield ($H_2O_2$%) and average electron transfer number ($n$) were calculated based on Eq. 1 and Eq. 2[45].

$$H_2O_2\% = \frac{2I_R/N}{I_D + (I_R/N)} \times 100\% \qquad (1)$$

$$n = 4 \times \frac{I_D}{I_D + (I_R/N)} \qquad (2)$$

where $I_D$ and $I_R$ are the background-corrected disk and ring currents, and $N$ is the calculated collect coefficient.

The electron transfer number can also be obtained by the Koutecký–Levich (K-L) equation in Eq. 3 and Eq. 4[46].

$$\frac{1}{i} = \frac{1}{i_d} + \frac{1}{i_k} = \frac{1}{Bw^{1/2}} + \frac{1}{i_k} \qquad (3)$$

$$B = 0.2nFc_0D_0^{2/3}\upsilon^{-1/6} \qquad (4)$$

where $i$, $i_d$ and $i_k$ are the collected current density, c density and kinetic current density, $\omega$ is the rotating rate of the electrode, $n$ is the average electron transfer number, $F$ is the Faraday constant, $c_O$ is the concentration of $O_2$, $D_O$ is the diffusion coefficients of $O_2$ and $\upsilon$ is the kinetic viscosity of the electrolyte.

Accelerated degradation tests (ADTs) were performed by the continuous cyclic voltammetry for 20000 cycles at the high potential range of 0.6 V and 1.0 V in the $O_2$-saturated $0.1\ mol\ L^{-1}$ HClO$_4$. The polarization curves as well as $H_2O_2$ selectivity before and after ADTs were compared to evaluate the stability of catalysts.

The sub-processes rate constants ($k_1$, $k_2$, $k_3$) for ORR were calculated by Damjanovic modeling[40]. The CV curves of CoIn-N-C were obtained by RRDE tests with catalyst loading of $120\ \mu g\ cm^{-2}$ in $O_2$ saturated $0.1\ mol\ L^{-1}$ HClO$_4$ at different rotation speeds of 400, 900, 1600, 2025, 2500 rpm. The main mathematical equations expressed as shown in Eqs. 5–7:

$$k_1 = S_2Z_1\frac{I_1N - 1}{I_1N + 1} \qquad (5)$$

$$k_2 = \frac{2Z_1S_2}{I_1N + 1} \qquad (6)$$

$$k_3 = \frac{NZ_2S_1}{I_1N + 1} \qquad (7)$$

Where $N$ is the calculated collect coefficient of RRDE, $D_{O2}$ and $D_{H2O2}$ are the diffusion coefficients of $O_2$ and $H_2O_2$ in $0.1\ mol\ L^{-1}$ HClO$_4$, respectively; The constants $Z_1$ and $Z_2$ are calculated by $Z_1 = 0.2D_{O_2}^{2/3}\upsilon^{-1/6}$ and $Z_2 = 0.2D_{H_2O_2}^{2/3}\upsilon^{-1/6}$ where $\upsilon$ is the kinetic viscosity. $I_1$ and $S_1$, $I_2$ and $S_2$ are the slopes and intercepts of linear fitting of $I_D/I_R$ vs $\omega^{-1/2}$ and $I_{DL}/(I_{DL} - I_D)$ vs $\omega^{-1/2}$, respectively, where $I_D$ and $I_R$ are the background-corrected disk and ring currents, $I_{DL}$ is the diffusion limited current, $\omega$ is the rotating rate of the electrode.

The $H_2O_2$ reduction performance was analyzed in the Ar-saturated $0.1\ mol\ L^{-1}$ HClO$_4$ solution with $10\ mmol\ L^{-1}$ $H_2O_2$.

The SCN$^-$ poisoning tests were performed by immersing the catalyst-coated RDE in $0.1\ mol\ L^{-1}$ KSCN for 30 s, and the ORR polarization curves were retested in $O_2$-saturated $0.1\ mol\ L^{-1}$ HClO$_4$.

### Electrochemical tests in flow cell

CoIn-N-C was sprayed onto the commercial gas-diffusion layer (GDL, AvCarb GDS 3250) with loading of $0.2\ mg\ cm^{-2}$ using the same catalysts ink described above.

For three-electrode tests, the anode and cathode tanks ($2 \times 1 \times 1\ cm^{-3}$ each) were separated by a pre-activated Nafion film, and $200\ mL$ $0.1\ mol\ L^{-1}$ HClO$_4$ as electrolyte was cycled at $50\ mL\ min^{-1}$ using pumps on each side. The CoIn-N-C/GDL (active area of $2 \times 1\ cm^2$), a platinum foil ($1 \times 1\ cm^2$), and reversible hydrogen electrode were used as the working electrode, counter electrode, and reference electrode, respectively. Pure $O_2$ was purged at $100\ mL\ min^{-1}$ though diffusion channel to the backside of the GDL during testing. The pH value of the electrolyte ($0.1\ mol\ L^{-1}$ HClO$_4$ $1.10 \pm 0.21$) was determined by a Hanna

HI504 pH controller (Hanna, Italy) calibrated with standard solutions. Solution resistances ($R$) were measured by electrochemical impedance spectroscopy at the corresponding working potential with 50 mV potential perturbation. All the measured potentials for flow cells were manually 100% iR compensated.

For two electrode tests, the commercial RuO$_2$ catalyst was sprayed onto the surface of pre-activated Nafion film with a Ru loading of $4\ mg_{Ru}\ cm^{-2}$, followed by hot-pressing at 135 °C and 2 MPa to obtain the anode. A piece of Ti mesh was utilized as anode current collector. $200\ mL$ $0.1\ mol\ L^{-1}$ HClO$_4$ as electrolyte was cycled at $50\ mL\ min^{-1}$ through cathode chamber ($0.1 \times 1 \times 1\ cm^{-3}$) and the anode using pumps on each side. Pure $O_2$ was purged at $100\ mL\ min^{-1}$ though diffusion channel to the backside of the GDL during testing. The polarization of the RuO$_2$ was evaluated ex-situ in a water electrolyzer, with Pt/C-coated GDE ($2\ mg_{Pt}\ cm^{-2}$) as cathode whose polarization is considered negligible comparing with OER. The internal resistance of the cell was evaluated by electrochemical impedance spectroscopy (EIS) from 0.1 to $10^6$ Hz with an amplitude of 0.05 V.

For chronopotentiometry test, various constant currents were applied by a CHI 760E electrochemistry station (CH Instrument) for 15 min, and for accumulation test, 200 mA was employed for 4 h. The concentration of $H_2O_2$ in the electrolyte on the cathodic side was determined by a traditional Ce(SO$_4$)$_2$ titration method based on the following equation:

$$2Ce^{4+} + H_2O_2 = 2Ce^{3+} + 2H^+ + O_2 \qquad (8)$$

As the orange Ce$^{4+}$ becomes colorless Ce$^{3+}$ upon reduction by $H_2O_2$, the concentration of Ce$^{4+}$ can be determined using ultraviolet-visible (UV-vis) adsorption spectrum. The calibration curve was obtained by measuring the UV-vis adsorption intensity at 316 nm of the standard Ce$^{4+}$ solution with known concentration.

The Faradic efficiency (FE) was calculated by Eq. 9

$$FE(\%) = \frac{2cVF}{It} \times 100\% \qquad (9)$$

where c and V are the calculated $H_2O_2$ concentration and the volume of the electrolyte from the cathodic tank, F is the Faraday constant, I and t are the operation current and time of the electrolysis test.

### Dye decomposition test

For the decomposition of organic dye, 1 mL ($1000\ mg\ L^{-1}$) FeSO$_4$ aqueous solution was added into 10 mL aqueous solution of methylene blue or rhodamine B ($60\ mg\ L^{-1}$), followed by 5 mL of the electrolyte from the cathodic tank after 30-min accumulation test. After thorough mixing, photos were taken based on the time sequence of 0 min, 2 min, 5 min, 10 min, 15 min. UV-vis adsorption spectra were also scanned at the same sequence.

### Computational method

We employed the Vienna Ab Initio Package (VASP) to perform all the density functional theory (DFT) calculations along with projected augmented wave (PAW) pseudopotentials[47,48] and the Perdew-Burke-Ernzerhof (PBE) functional[49]. The plane-wave cutoff energy was set to be 400 eV. The convergence criteria for self-consistent calculation were $10^{-5}$ eV energy difference, and 0.03 eV Å$^{-1}$ force change per atom. Grimme's DFT-D3 methodology[50] was used to describe the dispersion interactions. All structures were derived from a $6 \times 6$ supercell of graphene ($14.8 \times 14.8$ Å) periodicity in the x and y directions and one monolayer in the z direction by the vacuum depth of 20 Å in order to separate the surface slab from its periodic duplicates.

The 2e-ORR occurs as follows:

$$* + O_2 + H^+ + e^- = *OOH \tag{10}$$

$$*OOH + H^+ + e^- = H_2O_2 + * \tag{11}$$

where * represents the catalyst.

The change in Gibbs free energy ($\Delta G$) for the two reactions was calculated by Eq. 12:

$$\Delta G = \Delta E + \Delta ZPE - T\Delta S - neU \tag{12}$$

where $\Delta E$ is the change in calculated energies from DFT; $\Delta ZPE$ and $\Delta S$ are the change in zero-point energy and entropy obtained from vibrational frequencies; $T$ is 298.15 K; $n$ represents the number of transferred electrons, and $U$ is the electrochemical potential.

As indicated by the computational hydrogen electrode proposed by Nøskov, "$H^+ + e^-$" was assumed to be in equilibrium with $1/2$ $H_2$. The limiting potential ($U_L$) was defined as the lowest potential at which all the reaction steps were downhill in free energy, and can be calculated by Eq. 13.

$$U_L = \min(-\Delta G)/ne \tag{13}$$

## Data availability

All data needed to evaluate the conclusions in the paper are presented in the paper or the Supplementary Information. Source data are provided with this paper.

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

## Acknowledgements

This work is financially supported by National Natural Science Foundation of China (Grant No. 51634003), Heilongjiang Touyan Innovation Team Program (HITTY-20190033) and China Postdoctoral Science Foundation (2022M720952).

## Author contributions

J.D. and G.H. contributed equally. G.H. and C.D. conceived the project and designed the experiments. J.D., W.Z., L.L.,Y.Y., Y.S., and G.H. perform the experimental study. G.H. and X.Z. performed the theoretical study. J.D., G.H., L.G., Z.W., Y.X., G.Y.,and C.D. wrote the manuscript with support from all authors.

## Competing interests

The authors declare no competing interests.
