## [Peer Review File · Nature Communications]

REVIEWER COMMENTS

Reviewer #1 (Remarks to the Author):

This work demonstrates the synthesis of a Co–In dual-atom catalyst, which exhibits record-high H₂O₂ electrosynthesis activity in an acidic electrolyte. In atom (In–OH moiety) adjacent to the Co site is suggested to modify the OOH binding ability of the Co site and to optimize the activity. The high activity was well translated to a more practical gas diffusion electrode-based reactor where a high current density of 100 mA cm⁻² is achieved.

Because noble metal-based catalysts have enabled the H₂O₂ electrosynthesis in acidic media, this work presents excellent improvement in electrocatalytic performance. However, this reviewer found some major issues, particularly regarding the active site structure. These must be fully addressed in the major revision process to meet the high standard of Nature Communications. Detailed comments are listed below.

1. The authors claim that OH is spontaneously adsorbed after the catalyst is soaked in the electrolyte. The resulting In–OH moiety changes the OOH binding property of the adjacent Co site. However, the formation/presence of In–OH is not experimentally proved.
2. X-ray absorption analysis results must be reinvestigated. For the K-edge absorption, the oxidation state of the metal center is mostly estimated from the position of the absorption edge instead of the whiteline intensity. Therefore, the oxidation states of Co and In appear to be ~2 and >3, respectively. Other characterizations are required to support this complementarily.
3. Structural information was gained from the EXAFS fitting analysis. The fitting error must be included.
4. Increased IR-free potential (decreased overpotential) at higher current density in Figure 5b indicates IR overcompensation. The authors should check this and replace the IR-free values reported in the manuscript.
5. The authors claim that the h-SnO₂ catalyst exhibits a selectivity of 99.99%. This value is questionably close to 100%. A report of such high value requires multiple tests to guarantee the reproducibility of the results.
6. The authors should specify the metal contents in the catalysts.
7. The authors should add the reference for the IFEFFIT software (J. Synchrotron Radiat. 12, 537 (2005)).
8. The authors should add the experimental details for the ADT tests.

Reviewer #2 (Remarks to the Author):

This paper describes a dual atom catalyst (DAC) containing atomic Co and In species as a novel catalyst for the two-electron oxygen reduction reaction in acidic media to synthesis hydrogen peroxide. The presence of In species may regulate the electronic structure of Co species, leading to an optimised adsorption of the OOH intermediate thereby improving the performance for the 2-e ORR towards the generation of H₂O₂. The material has not been reported for 2-e ORR so far, however, the theory, namely modifying the electronic structure of atomic Co species for enhanced H₂O₂ electrosynthesis, has been reported extensively during the past few years. Many catalyst materials based on atomic Co species has been studies for the 2-e ORR, leading to exceptional performances in H₂O₂ production in either alkaline or acidic media. Therefore, the novelty of this work is not sufficient for Nature Communications. Besides that, there exists quite a few of technical issues in this work (detailed as follows), which need to be addressed.

- 1) The recent advancements in theoretical understanding and material innovation have already led to the development of a series of efficient catalysts for the electrosynthesis of H₂O₂, either in acidic or alkaline media.
- 2) What is the origin of the OH groups that are bound with the M_p atoms in the acidic media?
- 3) By simply adding the two metal precursor into the solution to prepare ZIF-8, how can the DCA structure be formed? The formation mechanism should be elaborated herein.
- 4) It's not clear in this study how important are the CoIn pairs in affecting the H₂O₂ productivity. From the characterizations provided in this study, the majority of Co and In species are existing as the non-paired atomic sites, with only a small portion forming the atomic pairs. The enhancement effects may be originated from the existence of two distinctive atomic species in the composites, rather than the small amount of the atomic pairs.
- 5) From Figure 3b, a small peak around 2.4 Å can be observed that may be related to the possible Co-M pairs. In Figure 3e, the authors did not mention the exact peak for the In-M pairs, however, a small peak is seen near 2.6 Å. Therefore, with the Co-M distance of 2.4 Å and the In-M distance ~ 2.6 Å, how can the simulation give the Co-In bond length of 2.72 Å that is close to the results from the STEM images?
- 6) Figure 5b is confusing. The authors attributed the overpotentials to the internal resistance of the flow cell, and after iR correction, the applied overpotentials remain low even at 100 mA cm⁻². The first question is, as seen from Figure 3b, with the application of iR correction, the overpotential decreases with the increase of current density (from 80 mA cm⁻² to 100 mA cm⁻²). How to explain this? Besides, to reach a current density of 100 mA cm⁻², a very negative voltage (without iR correction) of -3.5 V is required, under such a negative voltage, is ORR still the main reaction? Did the authors observe significant evolution of hydrogen during the reaction?
- 7) From Figure 4b, it can see the two-electron selectivity of the CoIn-N-C catalyst decrease concomitantly with the application of more negative potentials, from 94% at 0.60 V to merely ~ 80% at 0.40 V (vs. RHE). However, during the bulk production process, in the flow cell type reactor, the

selectivity is apparently much higher than that obtained with RRDE. For instance, at a current density of 80 mA cm⁻², the applied potential (after iR correction) is around -0.2 V (vs. RHE), while the FE for H₂O₂ is ~ 88%, which contradicts with the trend observed in the RRDE measurement.

Reviewer #3 (Remarks to the Author):

The manuscript reports CoIn dual-atom catalyst as a highly active catalyst for hydrogen peroxide electrosynthesis via 2e⁻ oxygen reduction reaction. From theory guided experiments, high activity of the dual atom catalyst resulting from the modulation effect of adjacent In atom was demonstrated, and characterization of dual-atom catalyst is well established. However, the modulation effect of dual atomic configuration predicted from DFT calculation results was not sufficiently supported by the experimental results. Therefore, this manuscript is not appropriate for publication, and detailed comments are listed below.

1. The mass ratio of the metal for both CoIn dual-atom and Co single atom which is critical in catalytic performance has not been provided in the manuscript. Therefore, it is necessary to conduct ICP analysis or XPS to quantification of this information.
2. The DFT calculation results was not sufficiently supported by the experimental results. Although the electronic modulation effect by adjacent M_pOH moieties is presented computationally, the modulation effect has not been experimentally demonstrated. It is necessary to analyze the changes in electron density of Co within the CoIn dual-atom catalyst as compared to Co in the Co single-atom catalyst. This analysis should be conducted using techniques such as X-ray absorption fine structure (XAFS) and X-ray photoelectron spectroscopy (XPS).
3. The characterization is insufficient to determine whether the enhanced activity of CoIn dual-atom catalyst results exclusively from the modulation of Co active site. Since there are many factors that affect the activity of 2e⁻ ORR activity, such as pore structure, oxygen functionality type and degree on carbon surfaces, metal loading density, and nitrogen content, it is necessary to verify whether these variables are adequately controlled.
4. To prove the high stability of the catalyst, post reaction characterization such as the presence of dual atom sites is required.
5. The manuscript reports high production rate in flow cell type reactor, but the overall cell voltage at high rate hindering practical applications cannot be overlooked. Therefore, it is recommended to demonstrate the production of H₂O₂ in a two-electrode flow cell system.
6. Typo: Fig 2g -> Fig 2f

Responses to Reviewers' Comments

(Manuscript: NCOMMS-22-53231)

Responses to Reviewer #1

Overall comment: *This work demonstrates the synthesis of a Co–In dual-atom catalyst, which exhibits record-high H₂O₂ electrosynthesis activity in an acidic electrolyte. In atom (In–OH moiety) adjacent to the Co site is suggested to modify the OOH binding ability of the Co site and to optimize the activity. The high activity was well translated to a more practical gas diffusion electrode-based reactor where a high current density of 100 mA cm⁻² is achieved. Because noble metal-based catalysts have enabled the H₂O₂ electrosynthesis in acidic media, this work presents excellent improvement in electrocatalytic performance. However, this reviewer found some major issues, particularly regarding the active site structure. These must be fully addressed in the major revision process to meet the high standard of Nature Communications. Detailed comments are listed below.*

Authors' reply: We are much grateful to the reviewer for his/her time and efforts devoted to evaluating our work and the positive comments. The issues proposed by the reviewer have been fully addressed as below.

Original Comment 1-1: *The authors claim that OH is spontaneously adsorbed after the catalyst is soaked in the electrolyte. The resulting In–OH moiety changes the OOH binding property of the adjacent Co site. However, the formation/presence of In–OH is not experimentally proved.*

Authors' reply: We thank the reviewer for this comment. In this revision, in situ surface-enhanced Raman scattering spectroscopy (SERS) was utilized to provide experimental evidence on the formation/presence of In–OH in acid solution.

As can be seen from Figure R1-1, when immersed to Ar-saturated HClO₄ electrolyte, a noticeable peak at ca. 1080 cm⁻¹ arises in the Raman spectra of CoIn–N–C comparing with that recorded in ambient air, which corresponds to the metal–OH bending mode, as reported in previous literatures(10.1038/s41560-018-0292-z, 10.1002/cey2.310). Due to the fact that similar peak can only be found in In–N–C, it is inferred that it is In atoms that adsorb OH in the electrolyte, which agrees well with the strong adsorption of OH on In atoms revealed by our calculation results.

Figure R1-1 Raman spectra of (a) CoIn-N-C, (b) In-N-C and (c) Co-N-C recorded in ambient air (ex-situ) and Ar saturated 0.1 M HClO₄ aqueous solution at different applied potential.

DFT calculation predicted that the adsorbed OH originated from H₂O. To provide experimental evidence, D₂O was utilized as solvent in replacement of H₂O. A noticeable peak appears at 735 cm⁻¹ while the peak at 1080 cm⁻¹ disappears in D₂O solution (Figure R1-2). Because the position shifting is in well consistence with the theoretical value calculated by the mass formula (equation R1-1), the peak at 735 cm⁻¹ is identified as In-OD, indicating that OH groups originate from H₂O. The production of OH species from dissociation of surface water is quite common on metals and metal oxides. Previous electron energy loss spectroscopy analysis (10.1016/j.susc.2015.10.026) strongly indicate that even at low temperatures, H₂O can dissociatively adsorb on indium oxide, yielding surface hydroxyls, which is also supported by DFT calculation (10.1021/jp801229g).

Figure R1-2 Raman spectra of CoIn-N-C recorded in ambient air (ex-situ) and Ar saturated 0.1 M HClO₄ solution in D₂O at different applied potential.

$$\begin{aligned}
 \gamma &= \nu(\text{OD}) / \nu(\text{OH}) \\
 &= \frac{\sqrt{m(\text{O}) + m(\text{D})}}{\sqrt{m(\text{O}) \times m(\text{D})}} / \frac{\sqrt{m(\text{O}) + m(\text{H})}}{\sqrt{m(\text{O}) \times m(\text{H})}} \quad (\text{Equation. R1-1}) \\
 &= \frac{\sqrt{16+2}}{\sqrt{16 \times 2}} / \frac{\sqrt{16+1}}{\sqrt{16 \times 1}} = 72.8\%
 \end{aligned}$$

Figure R1-1 and R1-2 have been adopted as Figure S13 in the revised SI, and the related descriptions about the Raman analysis have been added in the revised manuscript **on Page 10**. The experiment details have been added in the Method section **on Page 17** as well.

Original Comment 1-2: *X-ray absorption analysis results must be reinvestigated. For the K-edge absorption, the oxidation state of the metal center is mostly estimated from the position of the absorption edge instead of the whiteness intensity. Therefore, the oxidation states of Co and In appear to be ~2 and >3, respectively. Other characterizations are required to support this complementarily*

Authors' reply: We thank the reviewer for this comment. Because the Co K-edge spectra for our sample and all the references were collected by the transmission mode, both the edge position and the white line intensity can be utilized for the analysis of oxidation state. As suggested by the reviewer, the oxidation state of Co was analyzed by the position of the absorption edge (E_0 , defined by the energy of maxima in the 1st derivative of $\mu\chi(E)$) as well. According to Figure 1-3b, the E_0 of 7718.5 eV for CoIn-N-C indicates an oxidation state between 0 (Co foil with E_0 of 7709 eV) and +2 (CoO with E_0 of 7722.3 eV), which is consistent to the results obtained from white line intensity.

For In K-edge spectra, due to the poor signal noise ratio when collecting in transmission mode resulted from low concentration (1 wt%), the spectrum presented in the manuscript was collected by fluorescence mode. However, the data was processed incorrectly in logarithmic format, which is adopted only for data obtained from transmission mode, bringing deviation in both the edge position and white line intensity. We apologize for the mistake and the correct spectra are presented in Figure R1-3c. Due to the different collection mode between sample and references, only the edge position can be utilized for the oxidation state analysis. From the first derivative of $\mu\chi(E)$ in Figure R1-3d, the oxidation state of CoIn-N-C was identified as between 0 and +3.

Figure R1-3 (a) Co K-edge XANES and (b) their 1st derivatives for Co, CoO, CoPc, CoIn-N-C; (c) In K-edge XANES spectra and (d) their 1st derivatives for In, In₂O₃ and CoIn-N-C

As suggested by the reviewer, the oxidation state of Co and In was also analyzed by XPS. It is clear in Figure 1-4 that the binding energies for both Co 2p and In 3d orbitals are located between metal and oxides, which agrees well with XAFS analysis.

Related figures and descriptions have been revised in this revision, and the original conclusion about the oxidation state of Co and In remains unaffected.

Figure R1-4 (a) Co 2p and (b) In 3d XPS spectra for CoIn-N-C

Figure R1-3c has been adopted as Figure 3d in the revised manuscript. Figure 3e, 3f and 3h as well as Table S5 have been updated accordingly. Figure R1-3b and 1-3d have been adopted as Figure S10a and S10b in the revised SI. Descriptions about the oxidation state of Co and In have been updated on **Page 8-9**.

Original Comment 1-3: *Structural information was gained from the EXAFS fitting analysis. The fitting error must be included.*

Authors' reply: We thank the reviewer for this comment. The fitting results with error for CoIn-N-C is provided in Table R1-1.

Table R1-1 Fitting parameters for Co K-edge and In K-edge EXAFS for CoIn-N-C

	path	CN	R (Å)	σ^2 (10^{-3} Å)	ΔE_0 (eV)	R-factor
In K-edge	In-N ₁	2.00±0.03	2.3±0.05	7.0±1.6		
	In-N ₂	2.00±0.12	2.5±0.06	7.7±1.3	1.01±0.7	0.0031
	In-Co	1.00±0.11	2.7±0.11	2.1±1.3		
Co K-edge	Co-N ₁	1.99±0.11	1.9±0.12	8.3±0.5		
	Co-N ₂	2.00±0.15	2.0±0.19	9.5±1.6	1.65±0.9	0.013
	Co-In	0.92±0.32	2.7±0.14	5.0±2.5		

Table S5 has been updated with Table R1-1 in the revised SI.

Original Comment 1-4: *Increased IR-free potential (decreased overpotential) at higher current density in Figure 5b indicates IR overcompensation. The authors should check this and replace the IR-*

free values reported in the manuscript.

Authors' reply: We thank the reviewer for this comment. As suggested by the reviewer, the IR-drop was 80% compensated manually. However, the decreased overpotential at higher current density can still be seen (Figure R1-5a). It is noteworthy that similar phenomenon can also be seen in two-electrode system (cathode potential in Figure R1-6a). To summarize the results, we believe that the unusual phenomenon is not due to the IR overcompensation, **but due to shifted reaction selectivity**.

As can be seen in Figure R1-5b and R1-6b, the Faradic efficiency of H₂O₂ decreases at large current density, suggesting that the ORR selectivity partially shifted to 4e pathway, producing H₂O. It should be noted that the thermodynamic equilibrium potential of O₂/H₂O (1.23 V) is higher than that of O₂/H₂O₂ (0.69 V). This means that the cathode with higher selectivity to H₂O should produce a higher IR-free potential at the same applied current density, which is the case in our catalysts.

Figure R1-5 (a) Polarization curves for H₂O₂ electrosynthesis in three-phase flow cells with 80% iR-compensation and (b) H₂O₂ FE

Figure R1-6 (a) Analysis of polarization origin in two-electrode three-phase flow cells and (b) H₂O₂ FE

Figure R1-5b, 1-6a and 1-6b have been adopted as Figure 5b, 5g and S25b in the revised manuscript and SI, respectively. The details about the assembly and analysis of two electrode system have been added in Method section on **Page 19-20**.

Original Comment 1-5: *The authors claim that the h-SnO₂ catalyst exhibits a selectivity of 99.99%. This value is questionably close to 100%. A report of such high value requires multiple tests to guarantee the reproducibility of the results.*

Authors' reply: The mentioned h-SnO₂ with nearly 100% H₂O₂ selectivity was not the catalysts demonstrated in our work. We confirm that the performance in our manuscript is reproducible (Figure R1-7)

Figure R1-7 Multiple tests for RRDE curves and (b) H₂O₂ selectivity

Original Comment 1-6: 6. The authors should specify the metal contents in the catalysts.

Authors' reply: We thank the reviewer for this comment. According to ICP-OES, the metal contents in the catalysts are as follows.

Table R1-2 Metal contents in the catalysts detected by ICP-OES

Catalysts	Co wt. %	In wt. %
CoIn-N-C	0.9	1.05
Co-N-C	1.78	-
In-N-C	-	1.92

Table R1-2 has been adopted as Table S4 in the revised SI: Related description has been added on Page 7.

Original Comment 1-7: The authors should add the reference for the IFEFFIT software (*J. Synchrotron Radiat.* 12, 537 (2005))

Authors' reply: We thank the reviewer for this comment and have cited the related reference.

Original Comment 1-8: The authors should add the experimental details for the ADT tests.

Authors' reply: We thank the reviewer for this comment. The ADTs in RRDE were performed by the continuous cyclic voltammetry for 20000 cycles at the high potential range of 0.6 V and 1.0 V in the O₂-saturated 0.1 mol L⁻¹ HClO₄ at 1600 rpm. RRDE curves were collected before and after ADTs.

Before RRDE measurement, the Pt ring was cleaned by cyclic voltammetry between 0 V and 1.2 V for 50 cycles.

We have added related descriptions in the Method section in revised manuscript on **Page 19**.

Responses to Reviewer #2

Overall comment: *This paper describes a dual atom catalyst (DAC) containing atomic Co and In species as a novel catalyst for the two-electron oxygen reduction reaction in acidic media to synthesis hydrogen peroxide. The presence of In species may regulate the electronic structure of Co species, leading to an optimised adsorption of the OOH intermediate thereby improving the performance for the 2-e ORR towards the generation of H₂O₂. The material has not been reported for 2-e ORR so far, however, the theory, namely modifying the electronic structure of atomic Co species for enhanced H₂O₂ electrosynthesis, has been reported extensively during the past few years. Many catalyst materials based on atomic Co species has been studies for the 2-e ORR, leading to exceptional performances in H₂O₂ production in either alkaline or acidic media. Therefore, the novelty of this work is not sufficient for Nature Communications. Besides that, there exists quite a few of technical issues in this work (detailed as follows), which need to be addressed.*

Authors' reply: We are much grateful to the reviewer for the time and efforts devoted to the evaluation of our work. In this revision, with more experimental and theoretical evidence gained on active site structure, the in-situ evolution of coordination environment, the proposed mechanism of in-situ OH modification contributing to highly active dual atom sites for H₂O₂ electrosynthesis was fully supported, **which is firstly reported in ORR selectivity regulation**. The advancement of our catalysts to the performance of H₂O₂ electrosynthesis was discussed in our response to comment 2-1. We believe that our work is intensively improved to meet the standard of Nature Communications. Besides, the technical issues proposed by the reviewer have been fully addressed as below.

Original Comment 2-1: *The recent advancements in theoretical understanding and material innovation have already led to the development of a series of efficient catalysts for the, either in acidic or alkaline media*

Authors' reply: We thank the reviewer for this comment. Indeed, the development of efficient catalysts for H₂O₂ electrosynthesis is not rarely seen in recent years. However, the majority of reported catalysts to H₂O₂ electrosynthesis only show good performance in base solutions, and **those able to catalyze 2e ORR in practical acid media are far less active**, which is crucial in the possible industrial applications. As demonstrated in our manuscript, our CoIn-N-C is **the first catalyst able to produce**

a H_2O_2 partial current density of ca. 2 mA cm^{-2} (Figure 4e and Table S6), and when evaluated in flow cells, the production rate also exceeds other catalysts in acid (Table S8). The remarkable activity is further demonstrated in a practical two-electrode flow cell in this revision.

Besides enhanced reactivity, based on DFT prediction and in-situ characterization included in this revision, the **dual-metal site highly active in H_2O_2 electrosynthesis are formed after in-situ OH modification on In.** Different from traditional methodologies by regulating non-metal coordination environment, **the in-situ formation of In-OH is much more efficient in regulating the electron structure due to short-ranged metal-metal interaction.**

We have added related descriptions about the in-situ evidence and activity advancement in the revised manuscript. Please refer to our response to comment 2-2 and 2-6.

Original Comment 2-2: *What is the origin of the OH groups that are bound with the Mp atoms in the acidic media*

Authors' reply: We thank the reviewer for this comment. We conducted in-situ surface-enhanced Raman scattering spectroscopy (SERS) to study the origin of adsorbed OH in acid.

As can be seen from Figure R2-1a, when immersed to Ar-saturated HClO_4 electrolyte, a noticeable peak at ca. 1080 cm^{-1} arises in the Raman spectra of CoIn-N-C comparing with that recorded in ambient air, which corresponds to the metal-OH bending mode, as reported in previous literatures(10.1038/s41560-018-0292-z, 10.1002/cey2.310). When relacing H_2O with D_2O , it is clear that a noticeable peak appears at 735 cm^{-1} and the peak at 1080 cm^{-1} disappears (Figure R2-1b). Because the position shifting is in well consistence with the theoretical value calculated by the mass formula (Equation R2-1), the peak at 735 cm^{-1} is identified as In-OD. **Thus, it can be concluded that the adsorbed OH on In originate from H_2O in acid media.**

Figure R2-1 Raman spectra of CoIn-N-C recorded in ambient air (ex-situ) and Ar saturated 0.1 M HClO₄ (a) aqueous solution and (b) D₂O solution at different applied potential.

$$\begin{aligned} \gamma &= \nu(\text{OD}) / \nu(\text{OH}) \\ &= \frac{\sqrt{m(\text{O}) + m(\text{D})}}{\sqrt{m(\text{O}) \times m(\text{D})}} / \frac{\sqrt{m(\text{O}) + m(\text{H})}}{\sqrt{m(\text{O}) \times m(\text{H})}} \text{ (Equation. R2-1)} \\ &= \frac{\sqrt{16+2}}{\sqrt{16 \times 2}} / \frac{\sqrt{16+1}}{\sqrt{16 \times 1}} = 72.8\% \end{aligned}$$

Figure R2-1 has been adopted as Figure S13a and S13b in the revised SI, and the related descriptions about the Raman analysis have been added in the revised manuscript on **Page 10**. The experiment details have been added in the Method section on **Page 17** as well.

Original Comment 2-3: *By simply adding the two metal precursor into the solution to prepare ZIF-8, how can the DCA structure be formed? The formation mechanism should be elaborated herein.*

Authors' reply: We thank the reviewer for this comment.

Synthesizing ZIF-8 in the presence of metal salt followed by thermal treatment is one of the typical methods for the fabrication of DACs, which is widely adopted by researchers (10.1002/anie.202102053, 10.1021/jacs.9b08362, 10.1021/acscatal.1c02165).

We agree with the reviewer that uncovering the formation mechanism is an important aspect in the research of novel materials, including single atom catalysts (SACs) and DACs. However, due to the complexity of the thermal reaction system and the interference of high temperatures, the research on the formation process can be tough. For the relatively simple system of SACs, their formation by high-temperature thermal treatment has been a "black box" for a long time. It is not until recent years that progress has been made on simple and typical SACs systems, and three mechanisms have been proposed, including direct coordination inheritance from the precursors (eg. ZIF-8 to Zn SACs), ligand replacement (gaseous FeCl₃ & ZIF-8 to Fe SACs), and direct anchoring from metal species (Pd nanoparticles & ZIF-8 to Pd SACs). However, the understanding of the formation mechanism of more complicated systems including DACs via thermal reactions remains quite primary and is far from being uncovered.

To provide theoretical evidence on the preferential fixation of near-distanced CoIn pairs, the

formation energies (E_{form}) are calculated by DFT according to equation R2-2.

$$E_{form} = E_{CoIn} - E_{vac} - E_{Co} / 4 - E_{In} / 4 \quad (\text{Equation R2-2})$$

where E_{CoIn} and E_{vac} are energies of CoIn-embedded models and CoIn-free models with vacancies, E_{Co} and E_{In} are energies of a unit cell consisting of 4 Co or In atoms.

As shown in Figure R2-2, two models with middle (CoIn-m, $d_{Co-In}=4.9 \text{ \AA}$) and far (CoIn-f, $d_{Co-In}=7.4 \text{ \AA}$) CoIn distances have been utilized for comparison, while the model for CoIn DACs in the manuscript is denoted as CoIn-n. It is clear that the formation energies of CoIn models decrease with the shortened Co-In distance, suggesting that the fixation of near-distanced CoIn pairs is energetically preferential.

Figure R2-2 (a) Dual atomic models with short (CoIn), middle (CoIn-m) and far (CoIn-f) distances and (b) calculated formation energies.

Original Comment 2-4: *It's not clear in this study how important are the CoIn pairs in affecting the H2O2 productivity. From the characterizations provided in this study, the majority of Co and In species are existing as the non-paired atomic sites, with only a small portion forming the atomic pairs. The enhancement effects may be originated from the existence of two distinctive atomic species in the composites, rather than the small amount of the atomic pairs*

Authors' reply: We thank the reviewer for this comment. The amount of Co-In pairs are not small based on microscopic and spectroscopic evidence. As shown in the HAADF-STEM image in Figure 2d, the amount of Co-In pairs with distinct contrast are abundant throughout the field of view, similar to the conditions in images provided in Figure R2-3. Besides, the EXFAS can be well fitted by metal-N and Co-In paths, suggesting the existence of Co-In coordination is not rare (Figure 3 and Table S5). It is based on the abovementioned experimental evidence that Co-In pairs are proposed as a possible

active site for H₂O₂ electrosynthesis, which agrees well with our DFT prediction.

Figure R2-3 HAADF-STEM images and colored raster graphic of Co-In pairs

As suggested by the reviewer, there are indeed a few studies reporting catalysts with distinctive heteroatomic single atoms (10.1002/anie.202201007, 10.1002/adma.202003134, 10.1002/aenm.202203150, 10.1002/adfm.202210867), including some efficient catalysts to ORR. To exclude the possibility of distant Co-In atoms enhancing 2e ORR activity, we perform additional DFT calculations. Two models with middle (CoIn-m, $d_{\text{Co-In}}=4.9$ Å) and far (CoIn-f, $d_{\text{Co-In}}=7.4$ Å) CoIn distances have been analyzed. Similar to CoIn models presented in the manuscript, the adsorption of OH is strong on In atom in both models. The calculation of the reaction free energy diagram suggests that only short-ranged Co-In interaction is capable of enhancing H₂O₂ electrosynthesis activity (Figure R 2-4b and 4c).

Figure R2-4 (a) Dual atomic models with short (CoIn), middle (CoIn-m) and far (CoIn-f) distances and (b) calculated volcano plot and (c) free energy diagram.

Figure R2-4 has been adopted as Figure S16 in the revised SI, and the related descriptions have been added on **Page 11** in the revised manuscript.

Original Comment 2-5 : *From Figure 3b, a small peak around 2.4 Å can be observed that may be related to the possible Co-M pairs. In Figure 3e, the authors did not mention the exact peak for the In-M pairs, however, a small peak is seen near 2.6 Å. Therefore, with the Co-M distance of 2.4 Å and the In-M distance ~ 2.6 Å, how can the simulation give the Co-In bond length of 2.72 Å that is close to the results from the STEM images*

Authors' reply: We thank the reviewer for this comment. In fact, the Co-In bond length of 2.72 Å is obtained directly from the average of multiple measurements in STEM images (please refer to our response to comment 2-4).

For the XAFS fitting, it is noteworthy that the Frontier-transformed EXAFS spectra in Figure 3e and 3h are presented **without phase-correction**, which means that the visually observed peak centers

in the spectra do not directly correspond to the coordination length. The coordination length is given from in the best-fitting summarized in Table S5, where Co-In and In-Co of ca. 2.7 Å matches well with the microscopic observation.

Original Comment 2-6: *Figure 5b is confusing. The authors attributed the overpotentials to the internal resistance of the flow cell, and after iR correction, the applied overpotentials remain low even at 100 mA cm⁻². The first question is, as seen from Figure 3b, with the application of iR correction, the overpotential decreases with the increase of current density (from 80 mA cm⁻² to 100 mA cm⁻²). How to explain this? Besides, to reach a current density of 100 mA cm⁻², a very negative voltage (without iR correction) of -3.5 V is required, under such a negative voltage, is ORR still the main reaction? Did the authors observe significant evolution of hydrogen during the reaction?*

Authors' reply: We thank the reviewer for this comment.

1) About the polarization curve.

Authors' reply: We thank the reviewer for this comment. To exclude the possibility of over compensation, the IR-drop was 80% compensated manually. However, the decreased overpotential at higher current density can still be seen (Figure R2-5a). It is noteworthy that similar phenomenon can also be found in a two-electrode system (cathode potential in Figure 2-6a). Thus, we believe that the unusual phenomenon is not due to the IR overcompensation, but due to shifted reaction selectivity. As can be seen in Figure R2-5b and R2-6b, the Faradic efficiency of H₂O₂ decreases with the increasing current density, suggesting that the ORR selectivity partially shifted to 4e pathway, producing H₂O (considering minimum H₂ FE shown in Figure R2-7). It should be noted that the thermodynamic equilibrium potential of O₂/H₂O (1.23 V) is higher than that of O₂/H₂O₂ (0.69 V). This means that the cathode with higher selectivity to H₂O should produce a relatively higher IR-free potential at the same applied current density, which is the case in our catalysts at higher current densities.

Figure R2-5 (a) Polarization curves for H_2O_2 electrosynthesis in three-phase flow cells with 80% iR-compensation and (b) H_2O_2 FE

Figure R2-6 (a) Analysis of polarization origin in two-electrode three-phase flow cells and (b) H_2O_2 FE

2) About the H₂ FE

We understand the concern for H₂ evolution at low electrode potential, but because the cathode was fabricated based on gas-diffusion layer, no gas bubbles can be observed in the cathode during electrolysis. To reveal the reaction selectivity, the concentration of H₂ in the gas outlet was measured by gas chromatography, and the overall FE are summarized in Figure R2-7.

Figure R2-7 FE for H₂O₂ electrosynthesis in two-electrode three-phase flow cells

It is obvious that the production of H₂ is only minimum in flow cell. One of the reasons is the negligible activity of our CoIn catalysts to HER in the absence of O₂ (Figure R2-8). The other reason is that despite the much negatively shifted applied voltage, the actual cathode potential after iR-correction is not negative enough to trigger massive H₂ evolution (Figure R2-5a and R2-6a).

Figure R2-8 HER performance of CoIn-NC evaluated in Ar-saturated 0.1 M HClO₄

Figure R2-5b, R2-6a, R2-6b, R2-7 have been adopted as Figure 5b, 5g and S25b and S25a in the revised manuscript, and the related descriptions have been added on **Page 14 and 17** in the revised manuscript. The details about the assembly and analysis of two electrode system have been added on **Page 19-20** in Method section.

Original Comment 2-7: *From Figure 4b, it can see the two-electron selectivity of the CoIn-N-C catalyst decrease concomitantly with the application of more negative potentials, from 94% at 0.60 V to merely ~ 80% at 0.40 V (vs. RHE). However, during the bulk production process, in the flow cell type reactor, the selectivity is apparently much higher than that obtained with RRDE. For instance, at a current density of 80 mA cm⁻², the applied potential (after iR correction) is around -0.2 V (vs. RHE), while the FE for H₂O₂ is ~ 88%, which contradicts with the trend observed in the RRDE measurement.*

Authors' reply: We thank the reviewer for this comment. We believe that besides the applied potential, **the reactant supply in different testing conditions can also affect the selectivity of electrocatalytic reactions**, especially those involving gas reactant, as suggested in previous literature (10.1021/acscatal.1c03236).

In RRDE measurement, the reactant, O₂, diffuses within the electrolyte, finally reaching the electrode surface. The diffusion-limited current density (i_d) can be described by Levich equation (equation R2-3), suggesting that the diffusion current density has a maximum in a given electrolyte.

$$i_d = 0.2nFD^{2/3}\omega^{1/2}\nu^{-1/6}c \text{ (equation R2-3)}$$

where n is the electron transfer number, F is the Faraday constant, D is the diffusion coefficient, ω is the angular rotation rate of the electrode, ν is the kinematic viscosity and c is the concentration of O₂.

At low potential, with the increase of kinetic current, the reaction on catalyst-coated RRDE becomes diffusion controlled, which means that the concentration of O₂ drops to 0 at the electrode surface. In such circumstance, it is speculated that the surface O₂ is not sufficient, which will jeopardize the selectivity to H₂O₂. In flow cells, however, O₂ is supplied from behind the gas diffusion layer, and the O₂ supply is sufficient in the testing current density region.

To support our speculation, the H₂O₂ selectivity of CoIn-N-C-coated gas diffusion layer with different electrode conditions was tested in H-cell. As can be seen from Figure R2-9a, catalyst-coated commercial gas diffusion layer (GDL-hpho) is highly hydrophobic, and is capable of adsorbing a notable amount of gas when immersed to the electrolyte (Figure R2-9a). As a result, the O₂ supply is abundant, similar to the condition in flow cells. In the controlled group, the catalyst-coated gas diffusion layer was pre-treated in an alcoholic solution, which can transform the gas diffusion layer hydrophilic (GDL-hphi) (Figure R2-9a). In such circumstance, the O₂ is supplied through diffusion, similar to RRDE condition. It is obvious that at fixed electrode potential and active surface area, the

current produced by GDL-hpho is much higher than GDL-hphi, and the FE of H_2O_2 also exceeds (Figure R2-9b and 2-9c).

Figure R2-9 (a) Photographs, (b) the recorded current at fixed potential of 0.6V and (c) the H_2O_2 FE of catalyst-coated GDL-hpho and GDL-hphi

Responses to Reviewer #3

Overall comment: *The manuscript reports CoIn dual-atom catalyst as a highly active catalyst for hydrogen peroxide electrosynthesis via 2e- oxygen reduction reaction. From theory guided experiments, high activity of the dual atom catalyst resulting from the modulation effect of adjacent In atom was demonstrated, and characterization of dual-atom catalyst is well established. However, the modulation effect of dual atomic configuration predicted from DFT calculation results was not sufficiently supported by the experimental results. Therefore, this manuscript is not appropriate for publication, and detailed comments are listed below.*

Authors' reply: We are much grateful to the reviewer for the time and efforts devoted to the evaluation of our work. In this revision, we have gain more experimental and theoretical evidence on active site structure, the in-situ evolution of coordination environment and electrochemical performance in practical conditions. We believe that our work is intensively improved to meet the standard of Nature Communications.

Original Comment 3-1: *The mass ratio of the metal for both CoIn dual-atom and Co single atom which is critical in catalytic performance has not been provided in the manuscript. Therefore, it is necessary to conduct ICP analysis or XPS to quantification of this information.*

Authors' reply: We thank the reviewer for this comment. According to ICP-OES, the metal contents in the catalysts are as follows.

Table R3-1 Metal contents in the catalysts detected by ICP-OES

Catalysts	Co wt.%	In wt.%
CoIn-N-C	0.9	1.05
Co-N-C	1.78	
In-N-C		1.92

Table R3-1 has been adopted as Table S4 in the revised SI. Related description has been added on **Page 7**.

Original Comment 3-2: *The DFT calculation results was not sufficiently supported by the*

experimental results. Although the electronic modulation effect by adjacent $MpOH$ moieties is presented computationally, the modulation effect has not been experimentally demonstrated. It is necessary to analyze the changes in electron density of Co within the CoIn dual-atom catalyst as compared to Co in the Co single-atom catalyst. This analysis should be conducted using techniques such as X-ray absorption fine structure (XAFS) and X-ray photoelectron spectroscopy (XPS).

Authors' reply: We thank the reviewer for this comment. As suggest by the reviewer, additional XAS was performed on Co-N-C for comparison. It can be seen from the XANES in Figure R3-1a that the white line intensity of CoIn-N-C is slightly lower than that of Co-N-C and the XPS peaks for Co 2p shift to lower binding energy (Figure 3-1b). These results suggest that by introducing In, the oxidation state of Co slightly decreased, which is consistent with the smaller electronegativity of In (1.78) compared with Co (1.88).

Figure R3-1 (a) Co K-edge XANES and (b) Co 2p XPS spectra of Co-N-C and CoIn-N-C

In-situ surface-enhanced Raman scattering spectroscopy (SERS) was then conducted to gain experimental evidence on the adsorbed OH on In. As can be seen from Figure R3-2, when immersed to Ar-saturated $HClO_4$ electrolyte, a noticeable peak at ca. 1080 cm^{-1} arises in the Raman spectra of CoIn-N-C comparing with that recorded in ambient air, which corresponds to the metal-OH bending mode, as reported in previous literatures(10.1038/s41560-018-0292-z, 10.1002/cey2.310). Due to the fact that similar peak can only be found in In-N-C, it is inferred that it is In atoms that adsorb OH in the electrolyte, which agrees well with the strong adsorption of OH on In atoms revealed by our calculation results.

Figure R3-2 Raman spectra of (a) CoIn-N-C, (b) In-NC and (c) Co-NC recorded in ambient air (ex-situ) and Ar saturated 0.1 M HClO₄ aqueous solution at different applied potential.

Based on these results, it is believed that strong electron interaction exists within near-distanced Co-In pairs, and with the adsorption of OH on In in the electrolyte, further modulation can be expected.

Figure R3-1 and R3-2 have been adopted as Figure S11 and S13 in the revised SI, and the related descriptions about the oxidation state of Co as well as Raman measurement and analysis have been added in the revised manuscript on **Page 9-10**. The experiment details have been added in the Method section on **Page 17** as well.

Original Comment 3-3: *The characterization is insufficient to determine whether the enhanced activity of CoIn dual-atom catalyst results exclusively from the modulation of Co active site. Since there are many factors that affect the activity of 2e⁻ ORR activity, such as pore structure, oxygen functionality type and degree on carbon surfaces, metal loading density, and nitrogen content, it is*

necessary to verify whether these variables are adequately controlled.

Authors' reply: We thank the reviewer for this comment. Following the reviewer's suggestion, we have performed additional characterizations on Co-N-C and CoIn-N-C to study the physical properties of these two catalysts.

The N content detected by XPS is similar in Co-N-C and CoIn-N-C, and according to the fitting of N 1s XPS spectra, similar ratio of N chemical structure was revealed (Figure R3-4b and R3-4c). The ratio of pyrrolic N (less than 20%) considered to be efficient to 2e ORR in CoN_x moieties is only moderate compared with previous report (50 % in 10.1021/jacs.2c01194). These results exclude the possibility of shifted 2e ORR performance by N content or N configurations.

The surface O content studied by XPS is revealed to be similar (Figure R3-4a), and similar graphitic degree for both catalysts is suggested by the similar I_D/I_G ratio in Raman spectra (Figure R3-4d). These results exclude the effect of electrochemical performance by surface O functional groups or carbon defects.

N₂ adsorption/desorption analysis reveals similar specific surface area and pore structure for both samples (Figure R3-4e and R3-4f), with mesoporous structure commonly seen in ZIF-derived carbon materials.

Figure R3-4 (a) XPS survey spectra of Co-N-C, CoIn-N-C and In-N-C (b) N 1s XPS spectra and fitting results. (c) N content ratio, (d) Raman spectra of Co-N-C, In-N-C and CoIn-N-C. (e) N₂ adsorption/desorption

isotherms and (f) pore distribution curves

In summary, the contents and chemical environment of C, N, O and Co, carbon structure as well as pore structure are similar in Co-N-C and CoIn-N-C, leading to the conclusion that it is near-ranged Co-In pairs that are responsible for the remarkable 2e ORR performance.

Figure R3-4 have been adopted as Figure S15 in the revised SI, and the related descriptions about the activity origin have been added on **Page 11** in the revised manuscript.

Original Comment 3-4: *To prove the high stability of the catalyst, post reaction characterization such as the presence of dual atom sites is required.*

Authors' reply: We thank the reviewer for this comment. As suggested by the reviewer, we have performed HAADF-STEM characterization on CoIn-N-C after ADTs. It is revealed from Figure R3-5 that the Co-In pairs still exist after ADTs, suggesting high stability of CoIn pairs.

Figure R3-5 HAADF-STEM images and colored raster graphic of Co-In pair of catalysts after ADTs

DFT computations were also conducted to study the dissolution of metal species. The demetalization energies (E_{demet}) are calculated by DFT according to equation R3-2 and are utilized to support the stable fixation of near-distanced CoIn pairs.

$$E_{\text{demet}} = E_{\text{vac}} + E_{\text{Co}} / 4 + E_{\text{In}} / 4 - E_{\text{CoIn}} \quad (\text{Equation R3-2})$$

where E_{CoIn} and E_{vac} are energies of CoIn-embedded models and CoIn-free models with vacancies, E_{Co} and E_{In} (for dual-metal models only) are energies of a unit cell consisting of 4 Co or In atoms.

As shown in Figure R3-6, the demetalization energy for CoIn-n is the largest among three dual-metal models and exceeds that for Co single-atom model (s-Co). These results suggest that the metal pairs in CoIn-N-C are highly stable against demetalization.

Figure R3-6 (a) Dual atomic models with short (CoIn), middle (CoIn-m) and far (CoIn-f) distances; (b) illustration of demetalization and (c) calculated demetalization energies

Figure R3-5 and R3-6 have been adopted as Figure S18 and S19 in the revised SI, and the related descriptions about the catalyst stability have been added on **Page 12** in the revised manuscript.

Original Comment 3-5: *The manuscript reports high production rate in flow cell type reactor, but the overall cell voltage at high rate hindering practical applications cannot be overlooked. Therefore, it is recommended to demonstrate the production of H₂O₂ in a two-electrode flow cell system.*

Authors' reply: We thank the reviewer for this comment and agree with the reviewer that two-electrode flow cells are more practical for evaluating potential catalysts for industrial applications. As suggested by the reviewer, two-electrode flow cell with CoIn-N-C-coated carbon gas diffusion layer as cathode, and RuO₂-based anode was assembled and tested for H₂O₂ electrosynthesis.

In Figure R3-7, it is inspiring to notice that the cell voltage without *iR*-correction remains below 2.5 V even at high current of 140 mA (corresponding to a current density of ca. 100 mA cm⁻²), with selectivity to H₂O₂ of more than 80%. To study the origin of polarization, the potential of the cathode was evaluated ex-situ by the anodic potential (obtained by pairing the anode with Pt/C-coated cathode for hydrogen evolution reaction, HER, whose polarization is negligible comparing with OER) and *iR*-

drop in two-electrode system (Figure R3-7a). Noteworthy is that the cathode potential remains above 0.2 V at 140 mA (corresponding to ca. 100 mA cm⁻²), while the anode potential reaches 2.3 V. Considering the equilibrium potential of O₂/H₂O₂ (0.69 V) and O₂/H₂O (1.23 V), the cathode polarization (0.5 V) is not even half of that in the anode (1.1 V). As a result, the practicability of H₂O₂ electrosynthesis is not hindered by the cathode catalysts, and with advanced OER catalysts the cell voltage is expected to further decrease.

Figure R3-7 (a) Analysis of polarization origin in two-electrode three-phase flow cells (b) H₂O₂ FE

Figure R3-7a and 3-7b have been adopted as Figure 5g and S25 in the revised manuscript and SI. The details about the assembly and analysis of two electrode system have been added on **Page 14 and Page 19-20** in Method section.

Original Comment 3-6: *Typo: Fig 2g -> Fig 2f*

Authors' reply: We are sorry for this typo, which has been corrected in the revision.

REVIEWERS' COMMENTS

Reviewer #1 (Remarks to the Author):

The authors addressed the comments raised by this reviewer very well, and this paper can be published after one minor issue is resolved.

Regarding the Original Comment 1-3: The distance (R) should be expressed to the second decimal point because the error is also expressed to the second decimal point. Likewise, ΔE should be expressed to the first decimal point.

Reviewer #2 (Remarks to the Author):

The authors did perform a careful revision of the current work. Some of my concerns, such as the unusual fluctuation of the overpotentials upon the increase of current density and the origin of the OH groups absorbed on the metal sites, have been successfully addressed. The quality of this manuscript has been improved upon the revision. However, the novelty of this work is still not high enough for Nature Communications. Besides, the authors' claim over "the first catalyst able to produce a H₂O₂ partial current density of ca. 2 mA cm⁻²" is apparently not accurate. Many previous studies have already shown acidic H₂O₂ via ORR with current densities over 2 mA cm⁻² at much lower applied overpotentials. Therefore, I cannot suggest the acceptance of this work.

Reviewer #3 (Remarks to the Author):

The authors did diligent work to improve the reliability of the superior 2e⁻ ORR performance of CoIn-N-C catalyst. It seems that most of the comments have been addressed appropriately. In the revised manuscript, additional experiments and analyses have been included to provide supplementary evidence for the modulation effect of adjacent In atoms, as indicated by the computational results. Furthermore, controlled experiments make results more convincing by removing other variables that could affect the activity, such as surface structure and carbon support properties.

Therefore, this work can be published after minor improvements. To improve the comprehensibility of the performance of flow cells compared to half-cell results, it is recommended to provide additional experimental data. Specifically, to ensure a reliable comparison between the data of the three-phase flow cell presented in Figure 5a-c and the half-cell, it is recommended to incorporate a potential vs. current density graph, following the data represented in references such as (Nat. Catal. 2020, 3, 125–134, Nat Commun 2022, 13, 2668) to illustrate the data of the three-phase flow cell.

Responses to Reviewers' Comments

(Manuscript: NCOMMS-22-53231A)

Responses to Reviewer #1

Overall comment: The authors addressed the comments raised by this reviewer very well, and this paper can be published after one minor issue is resolved. Regarding the Original Comment 1-3: The distance (R) should be expressed to the second decimal point because the error is also expressed to the second decimal point. Likewise, ΔE should be expressed to the first decimal point.

Authors' reply: We thank the reviewer for the positive comment.

As suggested by the reviewer, modification on the significant digit in the EXAFS fitting results have been made. The distance (R) fitting results have been stabilized to the second decimal places and ΔE fitting results have been rounded to first effective decimal place. All the decimal places of the fitting results consist with their error, as provided in Table R1-1.

Table R1-1 Fitting parameters for Co and In K-edge EXAFS for CoIn -N-C

Sample	path	CN	R (Å)	σ^2 (10^{-3} Å)	ΔE_0 (eV)	R-factor
In K-edge	In-N ₁	2.00±0.03	2.30±0.05	7.0±1.6	1.0±0.7	0.003
	In-N ₂	2.00±0.12	2.50±0.06	7.7±1.3		
	In-Co	1.00±0.11	2.70±0.11	2.1±1.3		
Co K-edge	Co-N ₁	1.99±0.11	1.90±0.12	8.3±0.5	1.7±0.9	0.013
	Co-N ₂	2.00±0.15	2.00±0.19	9.5±1.6		
	Co-In	0.92±0.32	2.70±0.14	5.0±2.5		

ACTIONS: Table R1-1 has been adopted as Table S5 in the revised SI.

Responses to Reviewer #2

Overall comment: The authors did perform a careful revision of the current work. Some of my concerns, such as the unusual fluctuation of the overpotentials upon the increase of current density and the origin of the OH groups absorbed on the metal sites, have been successfully addressed. The quality of this manuscript has been improved

upon the revision. However, the novelty of this work is still not high enough for Nature Communications. Besides, the authors' claim over "the first catalyst able to produce a H₂O₂ partial current density of ca. 2 mA cm⁻²" is apparently not accurate. Many previous studies have already shown acidic H₂O₂ via ORR with current densities over 2 mA cm⁻² at much lower applied overpotentials. Therefore, I cannot suggest the acceptance of this work.

Authors' reply: We thank the reviewer for the time and efforts devoted in evaluating our manuscript. In this work, we have presented a strategy for catalyzing 2e ORR by the dual-atom catalyst (DAC) design. Unlike the traditional DACs for 4e ORR, where the dual metal sites can serve as the adsorption sites for O₂ via a bridge mode, Co is the sole adsorption site for O₂ in our CoIn DAC, due to the spontaneous and strong adsorption of OH on O-affinitive In atoms, as evidenced by in-situ characterizations, DFT computations as well as SCN⁻ poisoning tests. Interestingly, the end-on adsorption of O₂ enables a varied reaction pathway. This work widens the application of DACs to H₂O₂ electrosynthesis, and challenges the long-lasting idea that neighboring metal sites favor the complete reduction of O₂, as reported in J. Am. Chem. Soc. 2017, 139 (48), 17281–17284, Matter 2020, 3 (2), 509–521 and Nat. Commun. 2021, 12 (1), 1734. As a result of the electronic effect of OH-bonded In on Co atoms, the adsorption of O-containing intermediates is regulated, greatly shifting the reaction selectivity to 2e ORR. Because of the strong short-ranged metal-metal interaction between Co and In, the enhancement on ORR performance is more apparent than previous works.

Regarding the expression on electrochemical performance, we admit that there are many catalysts able to produce a partial H₂O₂ current more than 2 mA cm⁻² at the potential higher than 0.65 V in base solution, which, after a careful literature survey, cannot be achieved in acid media, as shown in Table R2-1. Due to the fact that the equilibrium potential of O₂/H₂O₂ is 0.7 V, producing a current density of 2 mA cm⁻² at the overpotential of only 50 mV is indeed a symbol of the activity of our CoIn DAC catalyst for H₂O₂ electrosynthesis.

Thus, the demonstration of dual metal catalysts in 2e ORR and the uncovering of the function mechanism, together with the electrochemical performance are the

significant strengths of this work. We believe that our paper is original and appealing to the readership of Nature communication.

Table R2-1 Comparison of the H₂O₂ production performance by RRDE

Samples	$V@2 \text{ mA cm}^{-2}$ (vs. RHE)	H ₂ O ₂ (%)	electrolyte	Ref.
CoIn-N-C	0.65	96	0.1 M HClO ₄	This work
PFC-72-Co	0.4	99	0.1 M HClO ₄	Nat. Commun.2022,1,13
CoN ₄ /VG	0.51	90	0.1 M HClO ₄	Energy Environ. Sci.2022,3,15
Pd ⁺ OCNT	0.14	92	0.1 M HClO ₄	Nat.Commun.2020,1,11
CB@Co-N-C	0.44	70	0.5 M H ₂ SO ₄	Adv. Funct. Mater.2023,27,33
Co NOC	0.42	88	0.1 M HClO ₄	Angew. Chem. Int. Ed. 2023,27,135
CoS ₂	0.06	60	0.05 M H ₂ SO ₄	ACS Catal.2019,9,9
O-CoSe ₂	0.3	80	0.05 M H ₂ SO ₄	Energy Environ. Sci.2020,11,13
Co-NC	0.425	90	0.1 M HClO ₄	J. Am. Chem. Soc.2022,32,144
Pd/NC	0.4	73	0.1 M HClO ₄	ACS Catal.2022,7,12
Co-NC/MXenes	0.3	78	0.5 M H ₂ SO ₄	Appl.Catal.B.2022,317,121737

Responses to Reviewer #3

Overall comment: The authors did diligent work to improve the reliability of the superior 2e⁻ ORR performance of CoIn-N-C catalyst. It seems that most of the comments have been addressed appropriately. In the revised manuscript, additional experiments and analyses have been included to provide supplementary evidence for the modulation effect of adjacent In atoms, as indicated by the computational results. Furthermore, controlled experiments make results more convincing by removing other variables that could affect the activity, such as surface structure and carbon support properties. Therefore, this work can be published after minor improvements.

To improve the comprehensibility of the performance of flow cells compared to

half-cell results, it is recommended to provide additional experimental data. Specifically, to ensure a reliable comparison between the data of the three-phase flow cell presented in Figure 5a-c and the half-cell, it is recommended to incorporate a potential vs. current density graph, following the data represented in references such as (Nat. Catal. 2020, 3, 125–134, Nat Commun 2022, 13, 2668) to illustrate the data of the three-phase flow cell.

Authors' reply: We thank the reviewer for the positive comment.

As suggested by the reviewer, we have provided the plot of potential vs. current density of CoIn-N-C catalyst, as shown in Figure R3-1, and have modified related descriptions in the manuscript.

Figure R3-1 Polarization curve for CoIn-N-C in three-electrode flow cell. The solution resistance (R_s) of $18.8 \pm 0.3 \Omega$ was determined by electrochemical impedance spectroscopy and the error represents the mean and standard deviation error of R_s measurements. iR correction can be achieved by subtracting the iR value from the measured potentials at applied each current density.

ACTIONS: Figure R3-1 has been adopted as Figure S25 in the revised SI:

Page 10: It is revealed that the H_2O_2 production rate ($9.68 \text{ mol g}^{-1} \text{ h}^{-1}$) with optimal FE can be achieved at 100 mA cm^{-2} (Figure 5b and Figure S25), comparable with that reported in base solution (Table S8).